
# Tropospheric temperature retrievals using nonlinear calibration functions in the pure rotational Raman lidar technique

Vladimir V. Zuev[1,2,3], Vladislav V. Gerasimov[1,2], Vladimir L. Pravdin[1], Aleksei V. Pavlinskiy[1], and Daria P. Nakhtigalova[1]

[1]Institute of Monitoring of Climatic and Ecological Systems SB RAS, Tomsk, 634055, Russia
[2]Tomsk State University, Tomsk, 634050, Russia
[3]Tomsk Polytechnic University, Tomsk, 634050, Russia

*Correspondence to*: Vladislav V. Gerasimov (gvvsnake@mail.ru)

**Abstract.** Among lidar techniques for temperature measurements, the pure rotational Raman (PRR) technique is the best-
10 suited for tropospheric and lower stratospheric temperature profiling. Calibration functions play a key role in the temperature retrieval algorithm from backscattered signals using the PRR lidar technique. The temperature retrieval accuracy and number of calibration coefficients depend on the selected calibration function. The commonly used calibration function linear in reciprocal temperature ignores the broadening of individual atmospheric $N_2$ and $O_2$ PRR lines and, at the same time, yields significant errors (±1 K) in temperature retrievals. However, the collisional (or pressure) broadening of $N_2$ and $O_2$ PRR lines
dominates over other types of broadening in the troposphere, and therefore, cannot be neglected during tropospheric temperature measurements. Gerasimov and Zuev (2016) derived mathematically a calibration function in the general analytical form that takes into account the collisional broadening of all $N_2$ and $O_2$ PRR lines. Nevertheless, this general calibration function represents an infinite series and cannot be directly used in the temperature retrieval algorithm. Therefore, four simplest nonlinear special cases (having three calibration coefficients) of the function, two of which have not been
suggested before, were considered and analyzed, and the best calibration function among them was determined via simulation. In this paper, we apply these special cases to real lidar remote sensing data, because all the functions take into account the collisional PRR lines broadening in varying degrees. The best-suited calibration function for tropospheric temperature retrievals is determined from the comparative analysis of temperature uncertainties yielded by using these functions. The absolute and relative statistical uncertainties of temperature retrieval are given in an analytical form assuming
Poisson statistics of photon counting. The vertical tropospheric temperature profiles, retrieved from nighttime lidar measurements in Tomsk (56.48° N, 85.05° E, Western Siberia, Russia) on 2 October 2014 and 1 April 2015, are presented as an example of the calibration functions application. The measurements were performed using a PRR lidar designed in the Institute of Monitoring of Climatic and Ecological Systems of the Siberian Branch of the Russian Academy of Sciences for tropospheric temperature measurements.



# 1 Introduction

The retrieval algorithm of a vertical temperature profile of the lower atmosphere from pure rotational Raman (PRR) raw lidar signals is known to consist of four main steps:

1. Smoothing PRR raw lidar signals and/or their ratio;
2. Lidar calibration, i.e. determination of the lidar calibration function coefficients by applying the least square method to the reference radiosonde (or model) data and previously smoothed lidar data;
3. Temperature profile retrieval by using the temperature retrieval function derived from the calibration function;
4. Estimation of the temperature retrieval absolute and relative uncertainties.

The PRR lidar technique suggested by Cooney (1972) is based on the temperature dependence of individual lines intensity of atmospheric $N_2$ and $O_2$ molecules PRR spectra. The intensity $I(T, \lambda)$ of a single PRR line of the wavelength $\lambda$ backscattered by excited $N_2$ or $O_2$ molecules can be expressed as (Penney et al., 1974)

$$I(\lambda, T) = PL\beta_\pi(\lambda, T), \tag{1}$$

where $P$ is the incident laser beam power; $L$ is the length of the scattering volume; $\beta_\pi(\lambda, T)$ is the backscatter cross section (or atmospheric backscatter coefficient). The backscattered signals of the Stokes and/or anti-Stokes branches of the spectra can be used for temperature determination. The intensities of individual PRR lines, corresponding to low and high rotational quantum numbers $J$ of the initial states of the PRR transitions, are of opposite temperature dependence (Behrendt, 2005). Namely, the intensity of each $N_2$ PRR line with $J_{low} \leq 8$ ($J_{low} \leq 9$ for $O_2$ PRR lines) decreases with increasing temperature, and conversely, the intensity of $N_2$ PRR lines with $J_{high} \geq 9$ ($J_{high} \geq 11$ for $O_2$ PRR lines) increases with increasing temperature, in both the branches of the spectra (Fig. 1). Note that only odd lines beginning with odd $J$ exist in $O_2$ molecule PRR spectrum (Wandinger, 2005). The ratio $Q(T)$ of backscattered signal intensities from two PRR-spectrum bands with opposite temperature dependence is required for air temperature $T$ determination. However, the PRR lidar theory (Cooney, 1972) gives the exact temperature dependence only for the intensity ratio of two individual PRR lines corresponding to certain $J_{low}$ and $J_{high}$

$$Q^{indiv.}(T) = \frac{I(J_{low}, T)}{I(J_{high}, T)} = \frac{\beta_\pi(J_{low}, T)}{\beta_\pi(J_{high}, T)} = \exp\left(\alpha + \frac{\beta}{T}\right), \tag{2}$$

where the constants $\alpha$ and $\beta$ are completely defined from the theory.

In practice, diffraction gratings (DGs) or interference filters (IFs) extract several adjacent PRR lines in the lidar temperature channels from backscattered light. IFs extract PRR lines from the anti-Stokes branches of $N_2$ and $O_2$ PRR spectra (Behrendt and Reichardt, 2000; Behrendt et al., 2002; Alpers et al., 2004; Di Girolamo et al., 2004; Radlach et al., 2008; Achtert et al., 2013; Newsom et al., 2013; Behrendt et al., 2015). DGs extract PRR lines from both the Stokes and



anti-Stokes branches of the spectra (Ansmann et al., 1999; Kim et al., 2001; Chen et al., 2011; Jia and Yi, 2014). Thus, one should consider the following expression (Arshinov et al., 1983)

$$Q^{\Sigma}(T) = \frac{I_{\text{low}}^{\Sigma}(T)}{I_{\text{high}}^{\Sigma}(T)} = \frac{\left[\sum_{J_{N_2}} \beta_{\pi}(J_{N_2}, T) + \sum_{J_{O_2}} \beta_{\pi}(J_{O_2}, T)\right]_{\text{low}}}{\left[\sum_{J_{N_2}} \beta_{\pi}(J_{N_2}, T) + \sum_{J_{O_2}} \beta_{\pi}(J_{O_2}, T)\right]_{\text{high}}}, \tag{3}$$

where $\beta_{\pi}(J_{N_2}, T)$ and $\beta_{\pi}(J_{O_2}, T)$ are the backscatter coefficients corresponding to $N_2$ and $O_2$ individual PRR lines,

respectively; $I_{\text{low}}^{\Sigma}(T)$ and $I_{\text{high}}^{\Sigma}(T)$ are the overall intensities of the PRR lines which enter the corresponding lidar temperature channels; indexes "low" and "high" show that summations in the numerator and denominator refer to the corresponding PRR-spectrum bands with $J_{\text{low}}$ and $J_{\text{high}}$. The ratio $Q^{\Sigma}(T)$ in Eq. (3) has a complicated temperature dependence and cannot be expressed as a simple function of $T$. Therefore, an approximation (or calibration) function $f_c^{\Sigma}(T)$ for the ratio $Q^{\Sigma}(T)$ is required to retrieve temperature profiles from lidar remote sensing data (Behrendt, 2005). The temperature retrieval

accuracy and the number of calibration coefficients depend on the selected calibration function.

Assuming that each PRR line profile represents the Dirac function, the general calibration function can be written in a natural logarithm form as follows (Gerasimov and Zuev, 2016)

$$\ln Q^{\Sigma}(T) \approx \ln f_c^{\Sigma}(T) = A + \frac{B}{T} + \frac{C}{T^2} + \frac{D}{T^3} + \cdots \quad \Leftrightarrow \quad y = A + Bx + Cx^2 + Dx^3 + \cdots, \tag{4}$$

where $A$, $B$, $C$, $D$, etc. are the calibration coefficients determined by applying the least square method to lidar remote sensing

(or simulation) data and reference radiosonde (or model) data; the symbol $\Leftrightarrow$ denotes the equivalence of expressions; $x = 1/T$ is the reciprocal temperature. The $n$-order in $x$ polynomial is assumed to retrieve temperature profiles with any desired accuracy depending on $n$ (Di Girolamo et al., 2004). The linear in $x$ special case of Eq. (4) with two calibration coefficients $A$ and $B$ (Arshinov et al., 1983) and the second-order in $x$ polynomial with three calibration coefficients $A$, $B$ and $C$ (Behrendt and Reichardt, 2000) are usually used by lidar researchers for temperature retrievals in the troposphere and lower

stratosphere. However, $N_2$ and $O_2$ PRR lines are broadened by the Doppler and molecular collision effects. Hence, their backscatter profiles are described by a Voigt function, which is a convolution of certain Gaussian and Lorentzian functions (Nedeljkovic et al., 1993). As the molecular collision effect dominates over the Doppler effect in the troposphere (Ivanova et al., 1993), one can consider the Lorentzian function for a PRR line shape description instead of the Voigt one (Ginzburg, 1972). Therefore, all collisionally broadened PRR lines contribute to the signals detected in both the lidar temperature

channels due to the long Lorentzian tails of the line profiles (Measures, 1984), and the general calibration function takes on the form (Gerasimov and Zuev, 2016)

$$\ln Q^{\text{all}}(T) = \cdots + \frac{A_{-2}}{T} + \frac{A_{-1}}{\sqrt{T}} + A_0 + A_1 \sqrt{T} + A_2 T + \cdots = \sum_{n=-\infty}^{\infty} A_n T^{\frac{n}{2}}, \tag{5}$$


where $A_n$ are the calibration coefficients and Eq. (4) represents a special case of Eq. (5). All the calibration functions mentioned above are valid only when the parasitic elastic signal backscattered by atmospheric aerosols and molecules is sufficiently suppressed in the lidar temperature channels. The state-of-the-art narrow-band IFs and DGs provide the suppression of the parasitic signal intensity in the channels up to 8–10 orders of magnitude (Achtert et al., 2013; Hammann and Behrendt, 2015; Hammann et al., 2015).

In order to take into account the atmospheric extinction of backscattered signals and their losses in the lidar transmitting and receiving optics, one should consider the lidar equation (Measures, 1984)

$$N(\lambda, z, T) = \eta N_0 G(\lambda, z) \frac{c\tau_0}{2} \xi(\lambda) \frac{A}{z^2} \beta_\pi(\lambda, z, T)\Theta^2(\lambda, z) , \qquad (6)$$

where $N(\lambda, z, T)$ is the number of backscattered photons (photocounts) detected by a photomultiplier tube (PMT) in a lidar temperature channel; $N_0$ is the number of emitted photons; $\eta$ is the PMT quantum efficiency; $G(\lambda, z)$ is the laser-beam receiver-field-of-view overlap; $\tau_0$ is the laser pulse duration; $c$ is the speed of light; $\xi(\lambda)$ is the transmittance of the lidar receiving optical system; $A$ is the receiver telescope area; $z$ is the scattering region altitude; and $\Theta(\lambda, z)$ is the transmission coefficient through the atmosphere between the scattering region and the lidar. Taking Eqs. (5) and (6) into account, the ratio of photocounts from two spectrally close bands involving several $N_2$ and $O_2$ PRR lines with $J_{low}$ and $J_{high}$ becomes (Newsom et al., 2012; Newsom et al., 2013)

$$Q(T, z) = \frac{N_{low}(T, z)}{N_{high}(T, z)} = \frac{G_{low}(z)}{G_{high}(z)} \exp\left( \sum_{n=-\infty}^{\infty} B_n T^{\frac{n}{2}} \right) = O(z) \exp\left( \sum_{n=-\infty}^{\infty} B_n T^{\frac{n}{2}} \right) , \qquad (7)$$

where $B_n$ are the calibration coefficients; $O(z)$ is the laser-beam receiver-field-of-view overlap function. At the complete overlap altitudes (usually above the atmospheric boundary layer), where $O(z) = 1$, Eq. (7) goes over into the calibration function like Eq. (5)

$$\ln Q(T) = \sum_{n=-\infty}^{\infty} B_n T^{\frac{n}{2}}. \qquad (8)$$

Note that the same result can be obtained on the assumption that the collisionally broadened elastic backscattered signal leaks into the nearest (to the laser line) lidar temperature channel (Gerasimov et al., 2015).

In our recent Optic Express paper, we considered the physics of our approach, derived mathematically the general calibration function that takes into account the collisional broadening of all $N_2$ and $O_2$ PRR lines, analyzed four nonlinear three-coefficient special cases of Eq. (8) via simulation to be used in the temperature retrieval algorithm, and determined the best function among them. In this paper, we apply these calibration functions to real lidar remote sensing data, because all the functions take into account the collisional PRR lines broadening in varying degrees, and determine the best-suited function for tropospheric temperature retrievals.



## 2 Special cases of the general calibration function

The general calibration function expressed by Eq. (8) represents an infinite series, and hence, the temperature retrieval function $T = T(Q)$ cannot be obtained in an analytical form from this infinite series. Therefore, one can use, e.g., some special cases of the integer power approximation of Eq. (8), i.e.

$$\ln Q(T) \approx \cdots + \frac{C_{-2}}{T^2} + \frac{C_{-1}}{T} + C_0 + C_1 T + C_2 T^2 + \cdots = \sum_{n=-\infty}^{\infty} C_n T^n, \qquad (9)$$

where $C_n$ are the calibration coefficients which can differ from $B_n$ in Eq. (8). Here we consider the linear (i.e. two-coefficient) and four simplest nonlinear (three-coefficient) in reciprocal temperature calibration functions and their corresponding temperature retrieval functions. Since Eq. (9) is a special case of Eq. (8), any special case of Eq. (9) represents automatically a special case of Eq. (8). The absolute and relative uncertainties of indirect temperature measurements are

10 obtained in an analytical form in Appendices A, A0–A4.

The frequently-used calibration function linear in $x = 1/T$ (Arshinov et al., 1983) is a special case of Eq. (9)

$$\ln Q = A_0 + \frac{B_0}{T} \iff y = A_0 + B_0 x, \qquad (10)$$

and its corresponding temperature retrieval function is

$$T = \frac{B_0}{\ln Q - A_0}, \qquad (11)$$

where $A_0$ and $B_0$ are the commonly designated calibration constants.

The most used nonlinear calibration function (Behrendt and Reichardt, 2000), containing the term quadratic in $x = 1/T$, also represents a special case of Eq. (9), i.e.

$$\ln Q = A_1 + \frac{B_1}{T} + \frac{C_1}{T^2} \iff y = A_1 + B_1 x + C_1 x^2, \qquad (12)$$

where $A_1$, $B_1$, and $C_1$ are the calibration constants. The corresponding temperature retrieval function is simply derived from

20 Eq. (12)

$$T = \frac{2C_1}{-B_1 + \sqrt{B_1^2 + 4C_1 \left( \ln Q - A_1 \right)}}. \qquad (13)$$

Another three-coefficient special case of Eq. (9) can be written as follows (Gerasimov and Zuev, 2016)

$$\ln Q = A_2 + \frac{B_2}{T} + C_2 T \iff y = A_2 + B_2 x + \frac{C_2}{x}, \qquad (14)$$





where $A_2$, $B_2$, and $C_2$ are the calibration constants. Solving Eq. (14), we have for the temperature retrieval function

$$T = \frac{2B_2}{(\ln Q - A_2) + \sqrt{(\ln Q - A_2)^2 - 4B_2 C_2}} . \qquad (15)$$

As it follows from the PRR lidar theory (Cooney, 1972), $y = \ln Q$ is a linear function of reciprocal temperature $x = 1/T$ (Arshinov et al., 1983). Conversely, the reciprocal temperature represents a linear function of $\ln Q$, i.e. $x = a + by$. In order to
take nonlinear effects into account, we consider the function

$$x = a + by + cy^2 \iff \frac{1}{T} = a + b \ln Q + c(\ln Q)^2, \qquad (16)$$

where $a$, $b$, and $c$ are some constants. Thus, a temperature profile can simply be retrieved via

$$T = \left[ c(\ln Q)^2 + b \ln Q + a \right]^{-1} \qquad (17)$$

or

$$T = \frac{C_3}{(\ln Q)^2 + B_3 \ln Q + A_3} , \qquad (18)$$

where $A_3 = a/c$, $B_3 = b/c$, and $C_3 = 1/c$. Equation (18) was first applied to real lidar data by Lee III (2013). Note that Eq. (16) represents a special case of Eq. (8), as we showed in our 2016 paper.

There exists another way to represent collisional PRR lines broadening (and therefore, nonlinear effects). Adding a term hyperbolic in $y = \ln Q$ to the linear calibration function of the form $x = a + by$ gives

$$x = A_4 + B_4 y + \frac{C_4}{y} \iff \frac{1}{T} = A_4 + B_4 \ln Q + \frac{C_4}{\ln Q} , \qquad (19)$$

where $A_4$, $B_4$, and $C_4$ are the calibration constants. Solving Eq. (19) yields

$$T = \frac{1}{A_4 + B_4 \ln Q + (C_4 / \ln Q)} = \frac{\ln Q}{B_4 (\ln Q)^2 + A_4 \ln Q + C_4} . \qquad (20)$$

## 3 The IMCES lidar setup

The IMCES PRR lidar was developed in Institute of Monitoring of Climatic and Ecological Systems of the Siberian Branch
of the Russian Academy of Sciences (IMCES SB RAS) for nighttime tropospheric temperature measurements (Fig. 2). A frequency-tripled Nd:YAG laser operating at a wavelength of 354.67 nm with 105mJ pulse energy at a pulse repetition rate of 20 Hz is used as the lidar transmitter. The backscattered signals (photons) are collected by a prime-focus receiving



telescope with a mirror diameter of 0.5 m. The IMCES lidar optical layout is shown in Fig. 3. The selection of spectrum bands containing PRR lines with $J_{low}$ and $J_{high}$ from both the Stokes and anti-Stokes branches of $N_2$ and $O_2$ PRR-spectra (Fig. 1) is performed via a double-grating monochromator (DGM). The DGM design and arrangement of optical fibers connecting both DGM blocks are the same as suggested by Ansmann et al. (1999). The main technical parameters of the

IMCES lidar transmitting, receiving, and data acquisition systems are summarized in Table 1. The spectral selection parameters of the DGM channels are listed in Table 2.

## 4 Temperature measurement example (1 April 2015)

In this section we consider an example of nighttime tropospheric temperature measurements performed with the IMCES lidar on 1 April 2015 in Tomsk (56.48° N, 85.05° E, Western Siberia, Russia). The lidar data were taken from 03:45 to 05:15

LT (or 31 March, 21:45–23:15 UTC), i.e. within 90 min integration time (108,000 laser shots). In order to determine the best calibration function, we compare and analyze five vertical tropospheric temperature profiles retrieved from the lidar data using Eqs. (11), (13), (15), (18), and (20).

### 4.1 Raw lidar data smoothing

In order to improve the signal-to-noise ratio, the raw lidar data (photocounts $N_L$ and $N_H$ detected by PMTs in the DGM

channels) should be smoothed. We tested more than dozens of different data-smoothing methods including the equal-sized and variable sliding-window smoothing ones presented in various papers (Behrendt and Reichardt, 2000; Behrendt et al., 2002; Alpers et al., 2004; Di Girolamo et al., 2004; Radlach et al., 2008; Radlach, 2009; Jia and Yi, 2014). The optimal data-smoothing method for our lidar system was the following. The IMCES lidar raw data were vertically smoothed with a variable sliding average window (Appendix A). Having the initial 48m length ($\Delta z = 24$ m, $k = 1$, and $n = 3$ in Eq. A10) in the

lidar to 240m altitude range, the variable sliding window was increased above and below by 24 m for every 240 m increase in altitude (see Fig. 4a). Note that similar lidar-data-smoothing procedure was used, e.g., in (Lee III, 2013). Due to low power of the IMCES lidar laser, the smoothed signals ratio $Q = \overline{N_L} / \overline{N_H}$ was additionally slightly smoothed using the equal-sized sliding window ($k = 5$, and $n = 11$ in Eq. A10) to reduce signal statistical fluctuations, as shown in Fig. 4b (see also the Supplement). For any other lidar system, the best data-smoothing method can differ from the method we used.

### 4.2 Reference temperature points for the lidar calibration

One of the problems we face during temperature measurements is the following. Unfortunately, we do not have our own radiosondes, and therefore, we have no possibility to launch a radiosonde simultaneously with lidar remote sensing at the lidar site. The two nearest to Tomsk meteorological stations launching radiosondes twice a day are situated in Novosibirsk (55.02° N, 82.92° E) and Kolpashevo (58.32° N, 82.92° E). Both the towns are at a distance of more than 250 km from



Tomsk. Hence, we cannot directly use vertical temperature profiles from these radiosondes as reference data points, which are known to be required for PRR lidars calibration. Nevertheless, we solved this problem as follows. We retrieved several points over Tomsk with a temperature accuracy of 0.5 K and a vertical accuracy of 20 m using the 925, 850, 700, 500, 400, 300, 200, and 100 hPa constant pressure altitude charts (CPACs), which can be found on http://gpu.math.tsu.ru/maps/.

Several CPACs are presented in the Supplement as an example. Two temperature profiles from radiosondes, launched on 1 April 2015 at 06:00 LT (00:00 UTC) in Novosibirsk and Kolpashevo, together with temperature points over Tomsk retrieved from the CPACs are shown in Fig. 5. The radiosondes data can be found on the webpage http://weather.uwyo.edu/upperair/sounding.html?region=np of the University of Wyoming (Novosibirsk and Kolpashevo station numbers are 29634 and 29231, respectively).

**4.3 Temperature profiles retrieved with different calibration functions**

Here we compare nighttime temperature profiles retrieved using five calibration functions considered in Sect. 2 from the altitude where the laser-beam receiver-field-of-view overlap is complete (~3 km) to 13 km (i.e. slightly above the local tropopause). Figure 6 presents a tropospheric temperature profile retrieved using the temperature retrieval function (Eq. 11) derived from the standard linear calibration function (Eq. 10). The absolute statistical uncertainty $\overline{\Delta T}$ of temperature

retrieval is calculated by Eq. (A21), whereas the relative uncertainty $(\overline{\Delta T}/T)$ is calculated by Eq. (A22). The difference in modulus $\left|T_{\text{CPAC}}-T\right|$ between temperature values retrieved from the CPACs and IMCES lidar data is also presented in Fig. 6. The nearest radiosondes data are given for comparison. Figures 7–10 show temperature profiles retrieved using the temperature retrieval functions expressed by Eqs. (13), (15), (18), and (20), respectively. These functions are derived from the corresponding nonlinear calibration functions, i.e. Eqs. (12), (14), (16), and (19).

Comparing all five profiles, one can see that, despite the lowest values of both the statistical uncertainties in the 3–12km altitude region ($\overline{\Delta T} < 0.5$ K, $(\overline{\Delta T}/T) < 0.004$) yielded by using Eq. (11), the difference $\left|T_{\text{CPAC}}-T\right|$ can exceed 5.5 K (Fig. 6). For the nonlinear functions in the same altitude region, the maximum difference $\left|T_{\text{CPAC}}-T\right|$ is less than 2.2 K and 1 K when using Eq. (13) and Eq. (20), respectively, as seen from Figs. 7 and 10. Similarly, for both the uncertainties we have: $\overline{\Delta T} < 1.5$ K, $(\overline{\Delta T}/T) < 0.013$ when applying Eq. (13), and $\overline{\Delta T} < 0.7$ K, $(\overline{\Delta T}/T) < 0.006$ for Eq. (20). Note that the

tropopause is located near 11km altitude. Taking into account all three parameters $\overline{\Delta T}$, $(\overline{\Delta T}/T)$, and $\left|T_{\text{CPAC}}-T\right|$, we can conclude that Eqs. (13), (15), (18), and (20) retrieve the tropospheric temperature much better compared to Eq. (11). Moreover, the two best-suited functions for temperature retrievals, which yield the minimum uncertainties and $\left|T_{\text{CPAC}}-T\right|$ among considered, are presented by Eqs. (18) and (20).



## 5 Temperature measurement example (2 October 2014)

Let us consider another example of nighttime tropospheric temperature measurements performed with the IMCES lidar on 2 October 2014 in Tomsk. The lidar data were taken from 20:21 to 21:21 LT (13:21–14:21 UTC), i.e. within 60 min integration time (72,000 laser shots). The raw and smoothed IMCES lidar signals together with raw and smoothed signals

ratios are presented in Fig. 11. We also compare five temperature profiles retrieved using Eqs. (11), (13), (15), (18), and (20). The temperature retrieval algorithm is the same as was applied to the lidar data dated 1 April 2015. For the lidar calibration, we retrieved temperature points over Tomsk using the corresponding CPACs. Two temperature profiles from radiosondes, launched on 2 October 2014 at 19:00 LT (12:00 UTC) in Novosibirsk and Kolpashevo, are also given for comparison.

Figure 12 shows a temperature profile retrieved using Eq. (11). For this profile in the 3–12km altitude region we have: $\Delta\overline{T} < 0.7$ K, $(\overline{\Delta T / T}) < 0.006$, and $\left|T_{\mathrm{CPAC}} - T\right| < 6.5$ K. Figure 13 shows temperature profiles retrieved using Eqs. (13) and (18). The temperature profiles retrieved using Eqs. (15) and (20) are presented in Fig. 14. As seen from Figs. 13 and 14, $\Delta\overline{T}$ $< 1.6$ K, $(\overline{\Delta T / T}) < 0.014$, and $\left|T_{\mathrm{CPAC}} - T\right| < 3.0$ K when applying Eq. (13); and $\Delta\overline{T} < 0.8$ K, $(\overline{\Delta T / T}) < 0.008$, and $\left|T_{\mathrm{CPAC}} - T\right| < 1.8$ K for Eq. (20) in the 3–12km altitude region. The tropopause is located near 12.3km altitude. Comparing

pairwise all the retrieved profiles for both the examples, one can see that $\Delta\overline{T}$, $(\overline{\Delta T / T})$, and $\left|T_{\mathrm{CPAC}} - T\right|$ values in the case of the latter example are higher than that for the former one (Sect. 4.3). This is due to that the smaller number of laser shots (and, therefore, photocounts detected in both the DGM channels) leads to the higher absolute and relative statistical uncertainties, as seen from Eqs. (A12) and (A13) in Appendix A. The two best-suited functions for temperature retrievals are seen in Figs. 13 and 14 to be the same as in the previous example (1 April 2015). The large difference between radiosonde

and lidar data temperature values in 2 to 3 km altitude region (Figs. 6 and 12; see also Lee III, 2013) is, perhaps, due to the incomplete laser-beam receiver-field-of-view overlap in the region. We also cannot exclude that any of the nonlinear calibration functions, in a varying degree, is able to correct for this incomplete overlap in the atmospheric boundary layer.

     The calibration coefficients of all the calibration functions used in both the temperature measurement examples can be found in the Supplement.

## 6 Summary and outlook

We have considered and used the linear and four nonlinear (three-coefficient) in $x = 1/T$ calibration functions in the tropospheric temperature retrieval algorithm. The corresponding temperature retrieval functions were applied to the nighttime temperature measurement data obtained with the IMCES lidar on 2 October 2014 and 1 April 2015. We also have



derived and used the absolute $\Delta\overline{T}$ and relative $(\overline{\Delta T / T})$ statistical uncertainties of indirect temperature measurements in an analytical form (Appendices A, A0–A4).

The comparative analysis of three parameters $\Delta\overline{T}$, $(\overline{\Delta T / T})$, and $\left| T_{\text{CPAC}} - T \right|$ showed:

- the nonlinear functions expressed by Eqs. (13), (15), (18), and (20), which take into account the collisional PRR lines broadening in varying degrees, retrieve the tropospheric temperature much better compared to the linear function (Eq. 11);

- equations (18) and (20) give the almost equally best-suited functions for the tropospheric temperature retrievals (although, Eq. 20 is slightly better than Eq. 18);

- the function given by Eq. (18) is the best from both the practical (real lidar data) and theoretical (simulation) points
    of view (Gerasimov and Zuev, 2016).

As the best function for temperature retrievals can depend on a lidar system (e.g., based on DGs or IFs for PRR lines extracting), it is reasonable to check all the mentioned nonlinear functions against lidar data obtained with different lidar systems to determine the best function in each specific case. As the collisional broadening of PRR lines is the largest in the atmospheric boundary layer, the nonlinear calibrations functions should be applied instead of the linear one for temperature
retrievals, especially if using a coaxial lidar. Furthermore, the stability of the calibration functions coefficients during long-time lidar measurements is one of the crucial aspects in determination of the best function. Therefore, it would be a good thing to study the coefficients stability during a day, week, month, etc., as well as it was done in (Lee III, 2013) for the linear calibration function coefficients.

## Appendix A: Absolute and relative uncertainties of temperature retrieval

Each value $T$ of a temperature profile retrieved from raw lidar data is known to be within the confidence interval $[T - \Delta T; T + \Delta T]$. The absolute uncertainty $\Delta T$ of indirect temperature measurements is defined in the general form as

$$\Delta T = \sqrt{\left( \frac{\partial T}{\partial Q} \Delta Q \right)^2} = \left| \frac{\partial T}{\partial Q} \Delta Q \right| = \left| \frac{\partial T}{\partial Q} \right| \cdot |\Delta Q|, \tag{A1}$$

where the temperature retrieval function $T = T(Q)$ is formally derived from the general calibration function or its any special case (see Sect. 2); $Q = N_{\text{L}}/N_{\text{H}}$ is the ratio of photocounts detected in the lidar temperature channels with $J_{\text{low}}$ and $J_{\text{high}}$,
respectively. The ratio $Q$ represents a function of two variables $N_{\text{L}}$ and $N_{\text{H}}$. Therefore the uncertainty $\Delta Q$ is defined as

$$\Delta Q = \left( \frac{\partial Q}{\partial N_{\text{L}}} \right) \Delta N_{\text{L}} + \left( \frac{\partial Q}{\partial N_{\text{H}}} \right) \Delta N_{\text{H}}, \tag{A2}$$

where both the partial derivatives are defined as



$$\frac{\partial}{\partial N_\mathrm{L}}\left(\frac{N_\mathrm{L}}{N_\mathrm{H}}\right)=\frac{1}{N_\mathrm{H}}, \quad \frac{\partial}{\partial N_\mathrm{H}}\left(\frac{N_\mathrm{L}}{N_\mathrm{H}}\right)=-\frac{N_\mathrm{L}}{(N_\mathrm{H})^2}. \tag{A3}$$

Substituting Eqs. (A3) into Eq. (A2), we obtain

$$\Delta Q = \frac{\Delta N_\mathrm{L}}{N_\mathrm{H}}-\frac{N_\mathrm{L}}{(N_\mathrm{H})^2}\Delta N_\mathrm{H}=Q\left(\frac{\Delta N_\mathrm{L}}{N_\mathrm{L}}-\frac{\Delta N_\mathrm{H}}{N_\mathrm{H}}\right). \tag{A4}$$

Assuming Poisson statistics of photon counting, we have for the 1–σ uncertainties of the registered number of photocounts
in both cases (Behrendt, 2005)

$$\Delta N_\mathrm{L} = \sqrt{N_\mathrm{L}}, \quad \Delta N_\mathrm{H} = \sqrt{N_\mathrm{H}}. \tag{A5}$$

Substituting Eqs. (A5) into Eq. (A4) and taking into account that $Q > 0$ and $N_\mathrm{L} > N_\mathrm{H}$, we get

$$|\Delta Q| = Q\left(\frac{1}{\sqrt{N_\mathrm{H}}}-\frac{1}{\sqrt{N_\mathrm{L}}}\right). \tag{A6}$$

Substituting Eq. (A6) into Eq. (A1), we obtain for the absolute uncertainty in the general form

$$\Delta T = \left|\frac{\partial T}{\partial Q}\right|Q\left(\frac{1}{\sqrt{N_\mathrm{H}}}-\frac{1}{\sqrt{N_\mathrm{L}}}\right). \tag{A7}$$

One can rewrite Eq. (A7) in the alternative form (Behrendt, 2005)

$$\Delta T = \left|\frac{\partial T}{\partial Q}\right|Q\sqrt{\frac{1}{N_\mathrm{H}}+\frac{1}{N_\mathrm{L}}}. \tag{A8}$$

Consequently, the relative uncertainty ($\Delta T/T$) of indirect temperature measurements is simply derived from Eqs. (A1) and
(A8)

$$\left(\frac{\Delta T}{T}\right)=\frac{1}{T}\sqrt{\left(\frac{\partial T}{\partial Q}\Delta Q\right)^2}=\left|\frac{\partial T}{\partial Q}\right|\frac{Q}{T}\sqrt{\frac{1}{N_\mathrm{H}}+\frac{1}{N_\mathrm{L}}}. \tag{A9}$$

However, Eqs. (A7)–(A9) are valid only for unsmoothed (raw) lidar data $N_\mathrm{L}$ and $N_\mathrm{H}$. In practice, raw lidar data are
previously smoothed in order to improve the signal-to-noise ratio. One of the most used data-smoothing methods is the
equal-sized (or variable) sliding-window smoothing (Behrendt and Reichardt, 2000; Behrendt et al., 2002; Alpers et al.,
2004; Di Girolamo et al., 2004; Radlach et al., 2008; Radlach, 2009; Lee III, 2013). The smoothed data $\overline{N}(z)$ and their





variance $\overline{\mathrm{Var}}(z)$ are related with the corresponding unsmoothed data $N(z)$ and variance $\mathrm{Var}(z)$ as follows (El'nikov et al., 2008)

$$\overline{N}(z) = \frac{1}{n}\left[N(z - k\Delta z) + \cdots + N(z) + \cdots + N(z + k\Delta z)\right]$$
$$= \frac{1}{2k+1}\sum_{i=-k}^{k} N(z + i\Delta z), \tag{A10}$$

$$\overline{\mathrm{Var}}(z) = \mathrm{Var}(z)/n, \tag{A11}$$

where $\Delta z$ is the vertical resolution of raw lidar data (initial vertical resolution); $k$ is the number of data points on either side of the central point; $n = 2k + 1$ is the sliding average window size, i.e. the number of raw lidar data points determining the sliding average window length (Otnes and Enochson, 1978). As the variance decreases by $n$ times, the absolute uncertainty $\Delta\overline{N}(z)$ of smoothed data decreases by $\sqrt{n}$ times. Therefore, the absolute uncertainty of temperature retrieval from the smoothed lidar data (photocounts) $\overline{N}_{\mathrm{H}}$ and $\overline{N}_{\mathrm{L}}$ is

$$\Delta\overline{T} = \frac{\Delta T}{\sqrt{n}} = \left|\frac{\partial T}{\partial Q}\right| \frac{Q}{\sqrt{n}}\left(\frac{1}{\sqrt{\overline{N}_{\mathrm{H}}}} - \frac{1}{\sqrt{\overline{N}_{\mathrm{L}}}}\right), \tag{A12}$$

where $Q = \overline{N}_{\mathrm{L}}/\overline{N}_{\mathrm{H}}$ . In this case, the confidence interval of the retrieved temperature profile is $[T - \Delta\overline{T}; T + \Delta\overline{T}]$, and the relative uncertainty is given by

$$\left(\frac{\overline{\Delta T}}{T}\right) = \left|\frac{\partial T}{\partial Q}\right| \frac{Q}{T\sqrt{n}}\sqrt{\frac{1}{\overline{N}_{\mathrm{H}}} + \frac{1}{\overline{N}_{\mathrm{L}}}}. \tag{A13}$$

If the second-order smoothing procedure (smoothing the previously smoothed data) is required, one can use instead of

Eqs (A10) and (A11) the following (El'nikov et al., 2008)

$$\overline{\overline{N}}(z) = \frac{1}{2l+1}\sum_{j=-l}^{l} \overline{N}(z + j\Delta z), \tag{A14}$$

$$\overline{\overline{\mathrm{Var}}}(z) = \overline{\mathrm{Var}}(z)/m = \mathrm{Var}(z)/(nm), \tag{A15}$$

where $l$ is the number of data points on either side of the central point; $m = 2l + 1$ is the sliding average window size. Therefore, the confidence interval of a retrieved temperature profile is $[T - \Delta\overline{\overline{T}}; T + \Delta\overline{\overline{T}}]$, where $\Delta\overline{\overline{T}} = \Delta\overline{T}/\sqrt{m} = \Delta T/\sqrt{nm}$ .

The absolute and relative uncertainties of temperature retrieval from the doubly smoothed lidar data $\overline{\overline{N}}_{\mathrm{H}}$ and $\overline{\overline{N}}_{\mathrm{L}}$ (and for $Q = \overline{\overline{N}}_{\mathrm{L}}/\overline{\overline{N}}_{\mathrm{H}}$ ) are given by





$$\Delta\overline{\overline{T}} = \left|\frac{\partial T}{\partial Q}\right| \frac{Q}{\sqrt{n\,m}} \left(\frac{1}{\sqrt{\overline{\overline{N}}_{\mathrm{H}}}} - \frac{1}{\sqrt{\overline{\overline{N}}_{\mathrm{L}}}}\right), \tag{A16}$$

$$\left(\overline{\overline{\frac{\Delta T}{T}}}\right) = \left|\frac{\partial T}{\partial Q}\right| \frac{Q}{T\sqrt{n\,m}} \sqrt{\frac{1}{\overline{\overline{N}}_{\mathrm{H}}} + \frac{1}{\overline{\overline{N}}_{\mathrm{L}}}}. \tag{A17}$$

Since the window size $n$ (and/or $m$) varies with altitude $z$, both the uncertainties should be estimated separately for each altitude interval where $n = $ const (and/or $m = $ const).

## 5   Appendix A0: Linear calibration function

For definiteness, we use Eqs. (A12) and (A13) to derive the absolute and relative uncertainties in an analytical form.

In order to obtain both the uncertainties for the linear calibration function, let us differentiate the temperature retrieval function derived from Eq. (10), i.e. (see Sect. 2)

$$T = \frac{B_0}{\ln Q - A_0}. \tag{A18}$$

10   The first-order derivative of the function is

$$\frac{\partial T}{\partial Q} = -\frac{B_0}{Q(\ln Q - A_0)^2}. \tag{A19}$$

Substituting Eq. (A19) into Eq. (A12), for the absolute uncertainty we get

$$\Delta\overline{T} = \frac{|B_0|}{(\ln Q - A_0)^2 \sqrt{n}} \left(\frac{1}{\sqrt{\overline{N}_{\mathrm{H}}}} - \frac{1}{\sqrt{\overline{N}_{\mathrm{L}}}}\right). \tag{A20}$$

One can rewrite Eq. (A20) in more simple form by substituting the expression $\ln Q - A_0 = B_0/T$ derived from Eq. (A18)

15   $$\Delta\overline{T} = \frac{T^2}{|B_0|\sqrt{n}} \left(\frac{1}{\sqrt{\overline{N}_{\mathrm{H}}}} - \frac{1}{\sqrt{\overline{N}_{\mathrm{L}}}}\right). \tag{A21}$$

Consequently, substituting Eq. (A19) into Eq. (A13), for the relative uncertainty we have

$$\left(\overline{\frac{\Delta T}{T}}\right) = \frac{1}{|\ln Q - A_0|\sqrt{n}} \sqrt{\frac{1}{\overline{N}_{\mathrm{H}}} + \frac{1}{\overline{N}_{\mathrm{L}}}} = \frac{T}{|B_0|\sqrt{n}} \sqrt{\frac{1}{\overline{N}_{\mathrm{H}}} + \frac{1}{\overline{N}_{\mathrm{L}}}}. \tag{A22}$$

Note that Eq. (A22) can also be expressed via signal-to-noise ratio (Chen et al., 2011).



**Appendix A1: Calibration function quadratic in $x = 1/T$**

The temperature retrieval function derived from Eq. (12) is written as (see Sect. 2)

$$T = \frac{2C_1}{-B_1 \pm \sqrt{B_1^2 + 4C_1 \left( \ln Q - A_1 \right)}} . \tag{A23}$$

The sign "+" instead of "±" should be chosen in the denominator of Eq. (A23), if $Q = \overline{N_L} / \overline{N_H}$ . When applying Eq. (A23)

for temperature retrievals, one should take into account the constraint coming from the square root. Namely, the expression

under the square root should be nonnegative, i.e. $B_1^2 + 4C_1 \left[ \ln Q(z) - A_1 \right] \geq 0$ or $\ln Q(z) \leq (B_1^2 / 4C_1) - A_1$ . Hence, Eq. (A23)

can retrieve the temperature profile $T$ only at altitudes $z$ where this condition holds.

The first-order derivative of the function is

$$\frac{\partial T}{\partial Q} = \frac{-4C_1^2 \left[ -B_1 + \sqrt{B_1^2 + 4C_1 \left( \ln Q - A_1 \right)} \right]^{-2}}{Q \sqrt{B_1^2 + 4C_1 \left( \ln Q - A_1 \right)}} . \tag{A24}$$

It is clear that the expressions for both the absolute and relative uncertainties will be cumbersome and poorly adapted for use

after substitution of this derivative in Eqs. (A12) and (A13). However, Eq. (A24) can be put in a more convenient form by

substituting the expressions which follow from Eq. (A23)

$$-B_1 + \sqrt{B_1^2 + 4C_1 \left( \ln Q - A_1 \right)} = \frac{2C_1}{T} ,$$

$$\sqrt{B_1^2 + 4C_1 \left( \ln Q - A_1 \right)} = \frac{2C_1}{T} + B_1 . \tag{A25}$$

After substitution of Eqs. (A25) into Eq. (A24), we can write instead of Eq. (A24)

$$\frac{\partial T}{\partial Q} = \frac{-T^3}{Q(2C_1 + B_1 T)} . \tag{A26}$$

Substituting Eq. (A26) into Eqs. (A12) and (A13), we obtain correspondingly for the absolute and relative uncertainties

$$\Delta \overline{T} = \frac{T^3}{\left| 2C_1 + B_1 T \right| \sqrt{n}} \left( \frac{1}{\sqrt{\overline{N_H}}} - \frac{1}{\sqrt{\overline{N_L}}} \right) , \tag{A27}$$

$$\left( \overline{\frac{\Delta T}{T}} \right) = \frac{T^2}{\left| 2C_1 + B_1 T \right| \sqrt{n}} \sqrt{\frac{1}{\overline{N_H}} + \frac{1}{\overline{N_L}}} . \tag{A28}$$





**Appendix A2: Calibration function hyperbolic in $x = 1/T$**

The temperature retrieval function in the general form derived from Eq. (14) represents (see Sect. 2)

$$T = \frac{2B_2}{(\ln Q - A_2) \pm \sqrt{(\ln Q - A_2)^2 - 4B_2C_2}} . \tag{A29}$$

For the case of $Q = \overline{N_L} / \overline{N_H}$ , the sign "+" instead of "±" should also be chosen in the denominator of Eq. (A29). Note that

Eq. (A29) can retrieve the temperature $T$ only at altitudes $z$ where the following condition holds: $[\ln Q(z) - A_2]^2 - 4B_2C_2 \geq 0$

or  $\ln Q(z) \geq A_2 + 2\sqrt{B_2C_2}$   (with $B_2C_2 \geq 0$).

The derivative of the temperature retrieval function is

$$\frac{\partial T}{\partial Q} = \frac{2B_2}{Q\left[(\ln Q - A_2) + \sqrt{(\ln Q - A_2)^2 - 4B_2C_2}\right]^2}$$
$$\times \left[1 + \frac{\ln Q - A_2}{\sqrt{(\ln Q - A_2)^2 - 4B_2C_2}}\right] . \tag{A30}$$

Equation (A30) can be put in a more convenient form by substituting the expressions which follow from Eqs. (A29) and

(14), respectively

$$(\ln Q - A_2) + \sqrt{(\ln Q - A_2)^2 - 4B_2C_2} = 2B_2/T ,$$
$$\ln Q - A_2 = B_2/T + C_2T . \tag{A31}$$

After substitution of Eqs. (A31) into Eq. (A30), we get for the derivative

$$\frac{\partial T}{\partial Q} = \frac{T^2}{Q\left(B_2 - C_2T^2\right)} . \tag{A32}$$

Substituting Eq. (A32) into Eqs. (A12) and (A13), we obtain for both the uncertainties

$$\Delta \overline{T} = \frac{T^2}{\left|B_2 - C_2T^2\right|\sqrt{n}}\left(\frac{1}{\sqrt{\overline{N_H}}} - \frac{1}{\sqrt{\overline{N_L}}}\right) , \tag{A33}$$

$$\left(\frac{\overline{\Delta T}}{T}\right) = \frac{T}{\left|B_2 - C_2T^2\right|\sqrt{n}}\sqrt{\frac{1}{\overline{N_H}} + \frac{1}{\overline{N_L}}} . \tag{A34}$$



**Appendix A3: Calibration function quadratic in $y = \ln Q$**

The first-order derivative of the temperature retrieval function, obtained from Eq. (16) (see Sect. 2)

$$T = \frac{C_3}{(\ln Q)^2 + B_3 \ln Q + A_3},$$
(A35)

is simply expressed as

$$\frac{\partial T}{\partial Q} = \frac{-C_3(2\ln Q + B_3)}{Q\left[(\ln Q)^2 + B_3 \ln Q + A_3\right]^2}.$$
(A36)

Substituting Eq. (A36) into Eq. (A12), for the absolute uncertainty we get

$$\Delta \overline{T} = \frac{\left|C_3(2\ln Q + B_3)\right|}{\left[(\ln Q)^2 + B_3 \ln Q + A_3\right]^2 \sqrt{n}}\left(\frac{1}{\sqrt{\overline{N_H}}} - \frac{1}{\sqrt{\overline{N_L}}}\right).$$
(A37)

Using the expression derived from Eq. (A35), i.e.

$$(\ln Q)^2 + B_3 \ln Q + A_3 = C_3/T,$$
(A38)

10 for the relative uncertainty we obtain

$$\left(\frac{\overline{\Delta T}}{T}\right) = \left|\frac{2\ln Q + B_3}{(\ln Q)^2 + B_3 \ln Q + A_3}\right|\frac{1}{\sqrt{n}}\sqrt{\frac{1}{N_H} + \frac{1}{N_L}}.$$
(A39)

In order to estimate both the uncertainties, one can also use Eqs. (A37) and (A39) in a more simple form. Substituting Eq. (A38) in Eqs (A37) and (A39), we obtain the following equations containing both $\ln Q$ and retrieved temperature $T$:

$$\Delta \overline{T} = \left|\frac{2\ln Q + B_3}{C_3}\right|\frac{T^2}{\sqrt{n}}\left(\frac{1}{\sqrt{\overline{N_H}}} - \frac{1}{\sqrt{\overline{N_L}}}\right),$$
(A40)

$$\left(\frac{\overline{\Delta T}}{T}\right) = \left|\frac{2\ln Q + B_3}{C_3}\right|\frac{T}{\sqrt{n}}\sqrt{\frac{1}{\overline{N_H}} + \frac{1}{\overline{N_L}}}.$$
(A41)

**Appendix A4: Calibration function hyperbolic in $y = \ln Q$**

Tropospheric temperature profiles are mentioned in Sect. 2 can also be retrieved via the function

$$T = \frac{\ln Q}{B_4(\ln Q)^2 + A_4 \ln Q + C_4},$$
(A42)



which first-order derivative is defined as

$$\frac{\partial T}{\partial Q} = \frac{C_4 - B_4 (\ln Q)^2}{Q \left[ B_4 (\ln Q)^2 + A_4 \ln Q + C_4 \right]^2} . \tag{A43}$$

Substituting Eq. (A43) in Eq. (A12), we obtain the absolute uncertainty containing only $\ln Q$

$$\Delta \overline{T} = \frac{\left| C_4 - B_4 (\ln Q)^2 \right|}{\left[ B_4 (\ln Q)^2 + A_4 \ln Q + C_4 \right]^2 \sqrt{n}} \left( \frac{1}{\sqrt{\overline{N}_{\mathrm{H}}}} - \frac{1}{\sqrt{\overline{N}_{\mathrm{L}}}} \right) . \tag{A44}$$

Using the expression derived from Eq. (A42), i.e.

$$B_4 (\ln Q)^2 + A_4 \ln Q + C_4 = (\ln Q)/T , \tag{A45}$$

for the relative uncertainty we get

$$\left( \frac{\overline{\Delta T}}{T} \right) = \left| \frac{C_4 - B_4 (\ln Q)^2}{B_4 (\ln Q)^3 + A_4 (\ln Q)^2 + C_4 \ln Q} \right| \frac{1}{\sqrt{n}} \sqrt{\frac{1}{\overline{N}_{\mathrm{H}}} + \frac{1}{\overline{N}_{\mathrm{L}}}} . \tag{A46}$$

Similarly, using Eq. (A45), one can rewrite Eqs. (A44) and (A46) in a practically useful form:

$$\Delta \overline{T} = \left| \frac{C_4}{(\ln Q)^2} - B_4 \right| \frac{T^2}{\sqrt{n}} \left( \frac{1}{\sqrt{\overline{N}_{\mathrm{H}}}} - \frac{1}{\sqrt{\overline{N}_{\mathrm{L}}}} \right) , \tag{A47}$$

$$\left( \frac{\overline{\Delta T}}{T} \right) = \left| \frac{C_4}{(\ln Q)^2} - B_4 \right| \frac{T}{\sqrt{n}} \sqrt{\frac{1}{\overline{N}_{\mathrm{H}}} + \frac{1}{\overline{N}_{\mathrm{L}}}} . \tag{A48}$$

**Acknowledgements**

We thank Dr. S. M. Bobrovnikov for helpful discussions. This study was conducted in the framework of the Federal Targeted Programme «R&D in Priority Fields of S&T Complex of Russia for 2014–2020» in the Priority Field "Rational use
of natural resources" (contract No. 14.607.21.0030, unique identifier ASR RFMEFI60714X0030).

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



**Table 1.** Main technical parameters of the IMCES lidar transmitting, receiving, and data acquisition systems.

|  |  |
| --- | --- |
| **Transmitting system** | |
| *Laser* | |
| Type | Unseeded frequency-tripled Nd:YAG |
| Model | Solar LS LQ529B |
| Wavelength | 354.67 nm |
| Spectral line width | $\sim 1 \text{ cm}^{-1}$ |
| Pulse repetition rate | 20 Hz |
| Pulse energy | 105 mJ |
| Pulse duration | 13 ns |
| Beam divergence | 0.3 mrad |
| Expansion factor | 10 |
| **Receiving system** | |
| *Telescope* | |
| Type | Prime-focus |
| Receiving mirror diameter | 0.5 m |
| Focal length | 1.5 m |
| Field of view | 0.4 mrad |
| *Optical fibers* | |
| F0 input fiber diameter | 0.55 mm (FG 550 UER) |
| F1 output fiber diameter | 0.6 mm (FT 600 UMT) |
| FB intermediate fibers diameter | 0.6 mm (FT 600 UMT) |
| F2 and F3 output fibers diameter | 1.5 mm (FT 1.5 UMT) |
| **Double-grating monochromator** | |
| *Lens L1, L2* | |
| Diameter | 130 mm |
| Focal length | 300 mm |
| *Diffraction gratings DG1, DG2* | |
| Grooves / mm | 2100 |
| Diffraction order | 2 |
| Diffraction angle | 48.151° |



| Data acquisition system | |
| --- | --- |
| Photomultiplier tubes PMT1–PMT3 | Hamamatsu R7207-01 |
| PMTs quantum efficiency | 25% |
| Photon counter | PHCOUNT_4 (IMCES SB RAS) |
| Number of channels | 4 (3 in use) |
| Counting rate | Up to 200 counts/s |
| Initial vertical resolution | 24 m |





**Table 2.** Spectral selection parameters of the DGM channels (central wavelength (CWL) and full width at half maximum (FWHM)).

| DGM channel | CWL, nm | FWHM, nm/cm$^{-1}$ |
|---|---|---|
| $J_{low}$ (Stokes) | 355.22 | ~0.22/17 |
| $J_{low}$ (anti-Stokes) | 354.12 | ~0.22/17 |
| $J_{high}$ (Stokes) | 356.03 | ~0.35/28 |
| $J_{high}$ (anti-Stokes) | 353.32 | ~0.35/28 |





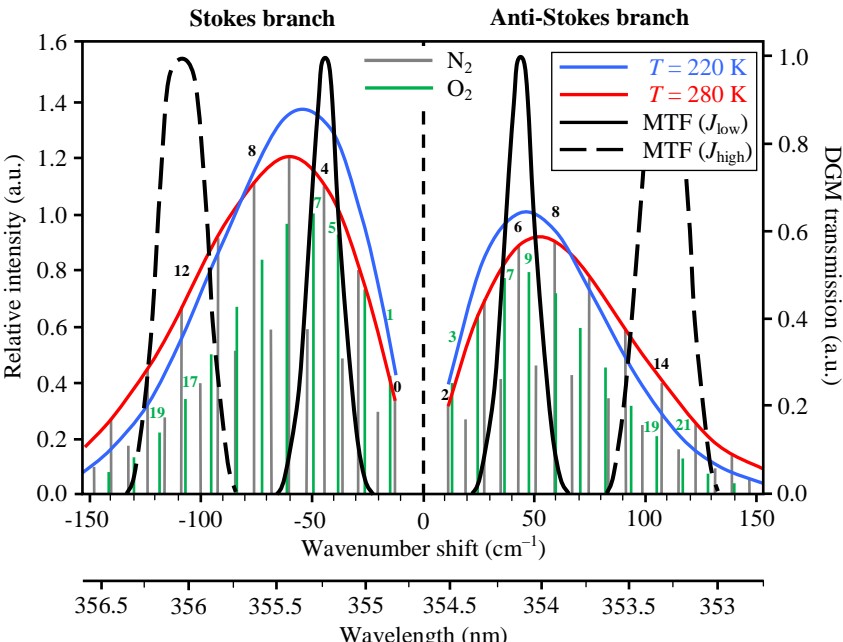

**Figure 1.** The equidistant PRR spectra of $N_2$ and $O_2$ linear molecules, schematic drawing of the IMCES lidar monochromator transmission functions (MTF) and envelopes of $N_2$ PRR spectrum at different temperatures. The index over a spectral line denotes the rotational quantum number $J$ of the initial state of the transition. The spectral line number and number $J$ are the same for the Stokes branch. All PRR lines intensities are normalized to the intensity of $N_2$ PRR line with $J = 6$ of the anti-Stokes branch at $T = 220$ K.

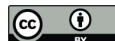



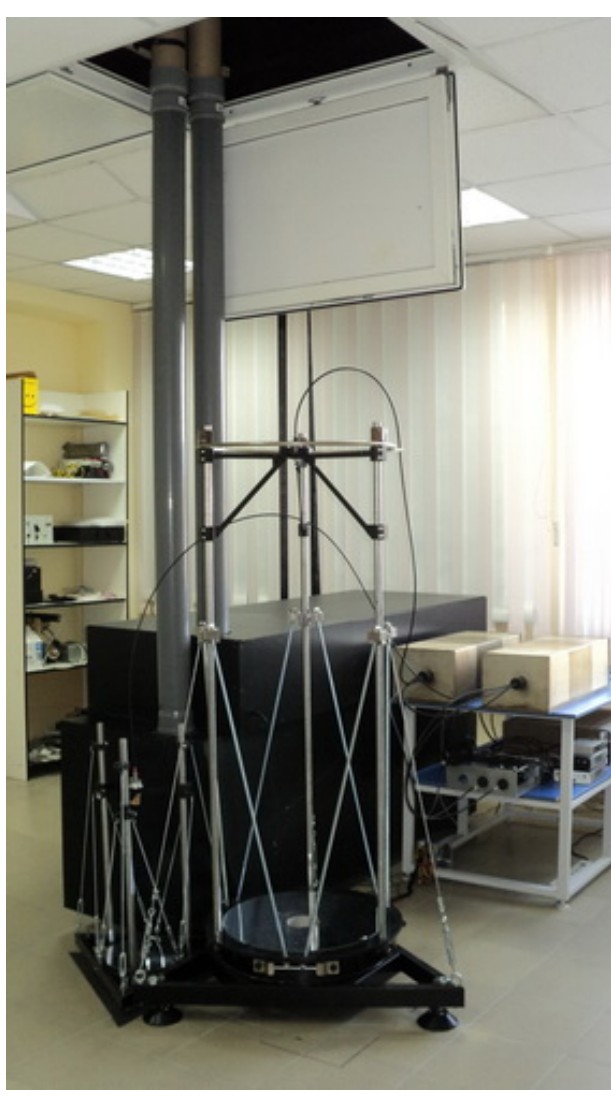

**Figure 2.** The IMCES PRR lidar.



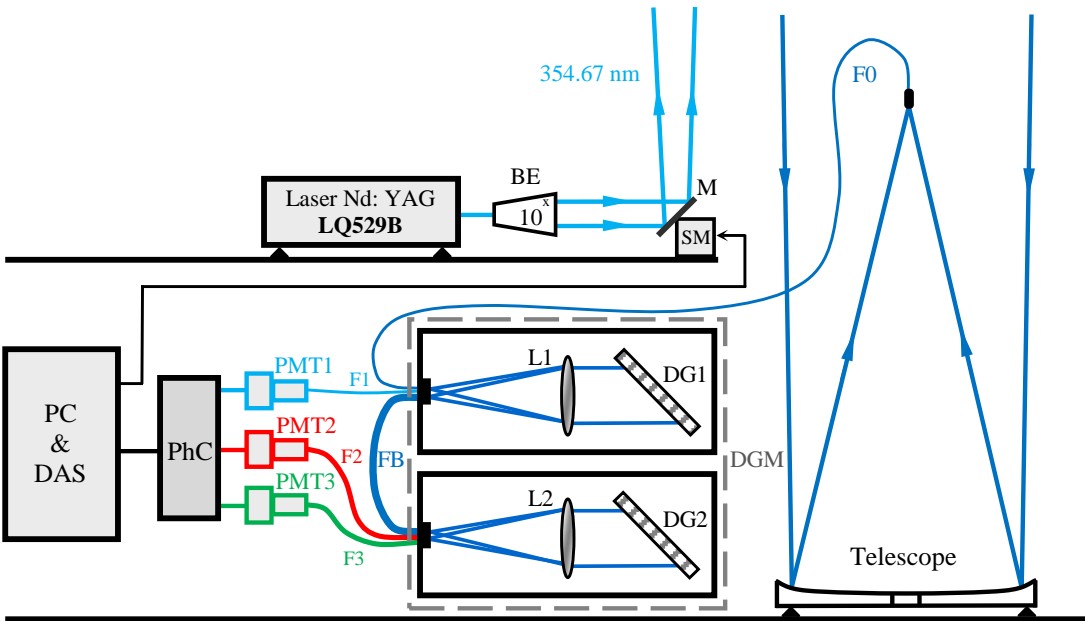

**Figure 3.** IMCES lidar optical layout: PC&DAS, personal computer and data acquisition system; PhC, photon counter; PMT1–PMT3, photomultiplier tubes; F0–F3, optical fibers; FB, four fiber bundle, connecting two monochromator blocks; DGM, double-grating monochromator; L1 and L2, lenses; DG1 and DG2, diffraction gratings; BE, beam expander with expansion factor of 10; M, mirror; SM, stepping motor.





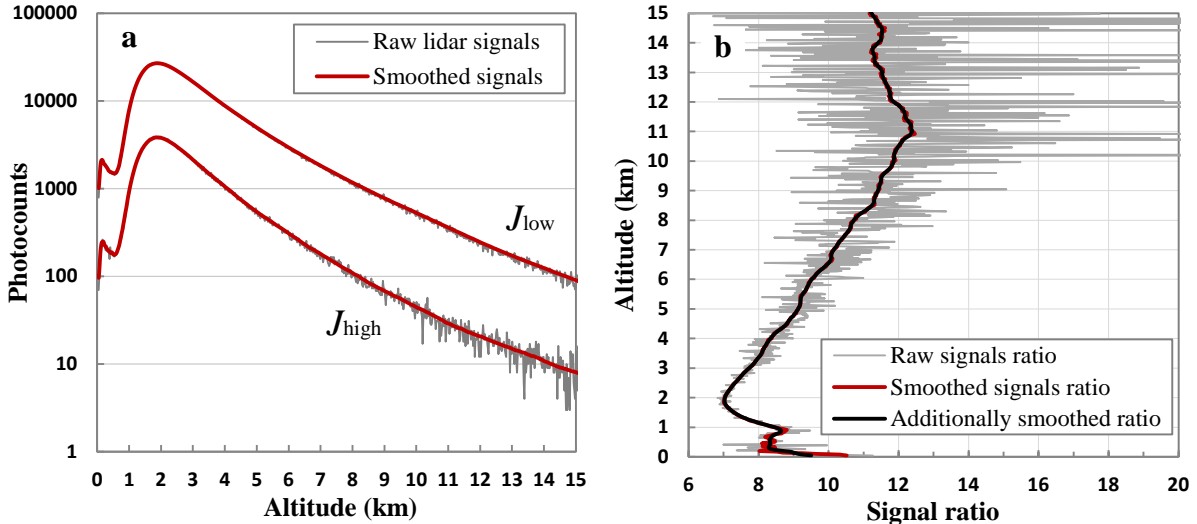

**Figure 4.** IMCES lidar data taken between 03:45 and 05:15 LT on 1 April 2015 (31 March, 21:45–23:15 UTC). **(a)** Raw photocounts $N_\mathrm{L}$ and $N_\mathrm{H}$ detected in the lidar channels with $J_\mathrm{low}$ and $J_\mathrm{high}$, respectively, together with the smoothed ones $\overline{N}_\mathrm{L}$ and $\overline{N}_\mathrm{H}$. **(b)** Raw photocounts ratio $Q = N_\mathrm{L}/N_\mathrm{H}$, smoothed photocounts ratio $Q = \overline{N}_\mathrm{L}\big/\overline{N}_\mathrm{H}$, and additionally smoothed ratio.





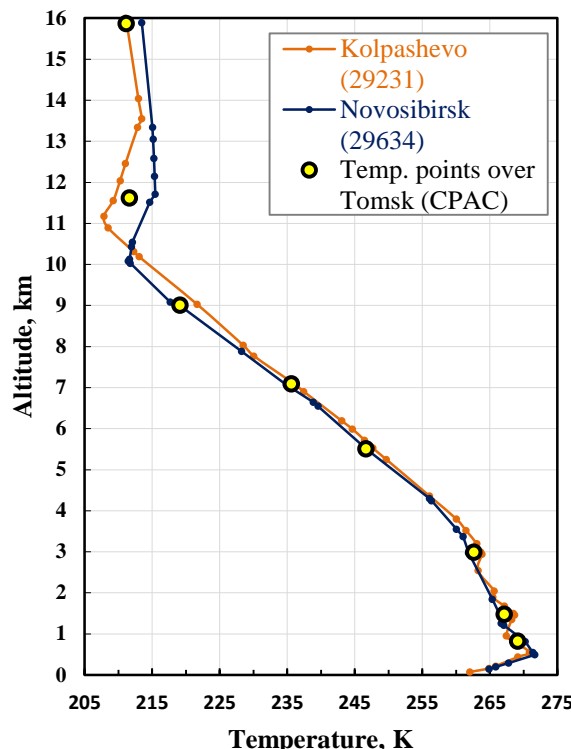

**Figure 5.** Temperature profiles from radiosondes launched on 1 April 2015 at 06:00 LT (00:00 UTC) in Novosibirsk (station 29634) and Kolpashevo (station 29231) as well as temperature points over Tomsk retrieved from the GISMETEO constant pressure altitude charts (CPACs).



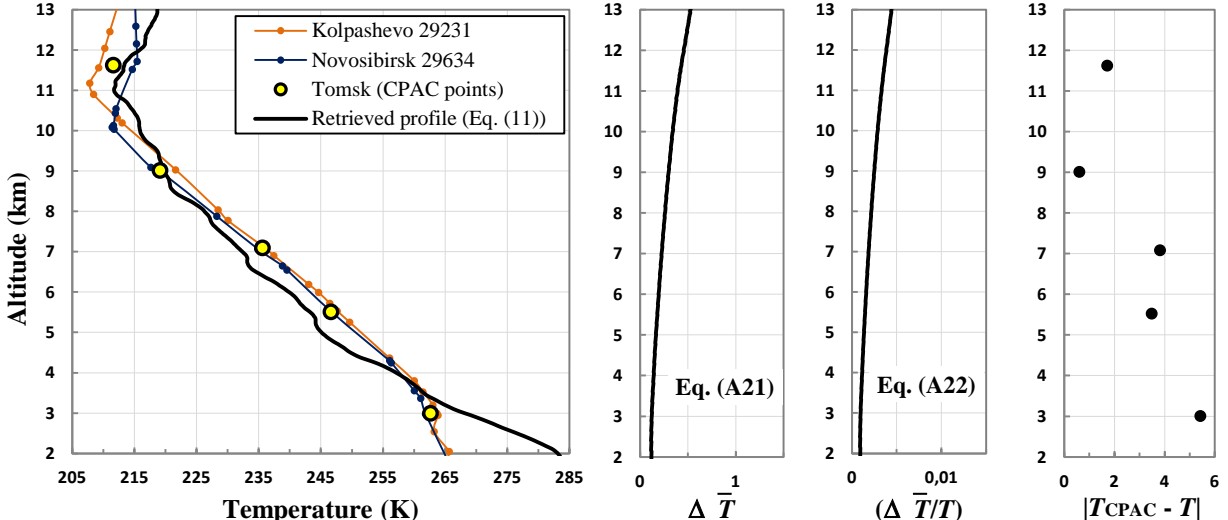

**Figure 6.** (1 April 2015) Temperature profile retrieved using the temperature retrieval function (Eq. 11) derived from the standard linear calibration function (Eq. 10, Arshinov et al., 1983). The absolute and relative uncertainties $\Delta\overline{T}$ and $(\overline{\Delta T / T})$ are calculated by Eqs. (A21) and (A22), respectively. The values $T_{\text{CPAC}}$ over Tomsk are retrieved from the 700, 500, 400, 300, and 200 hPa constant pressure altitude charts (CPACs). The radiosondes data from the nearest station in Novosibirsk and Kolpashevo are given for comparison.





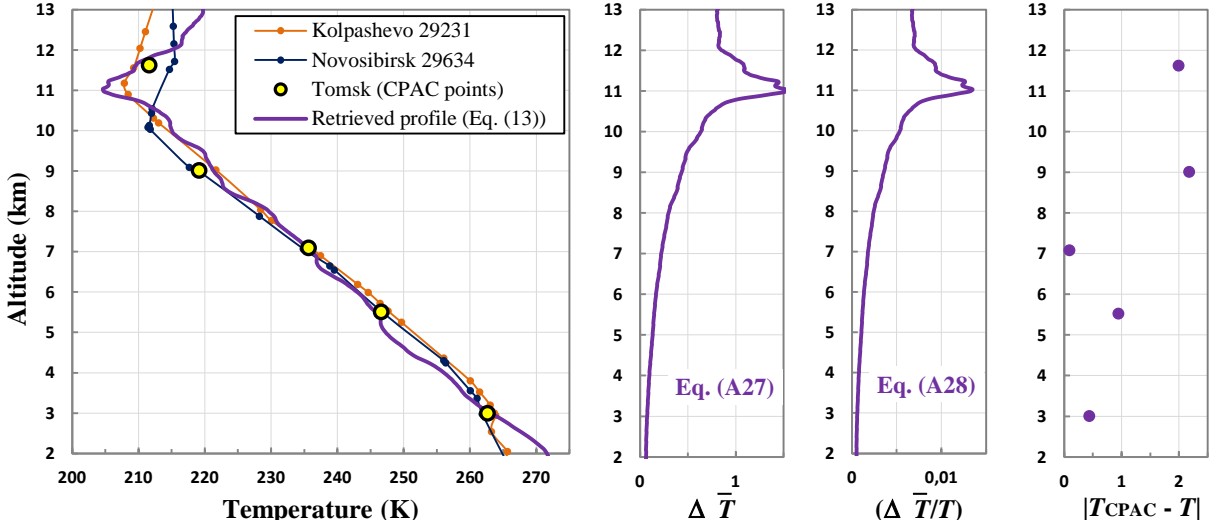

**Figure 7.** (1 April 2015) Temperature profile retrieved using the temperature retrieval function (Eq. 13) derived from the standard calibration function suggested by Behrendt and Reichardt (2000). The uncertainties $\Delta \overline{T}$ and $(\overline{\Delta T / T})$ are calculated by Eqs. (A27) and (A28), respectively.

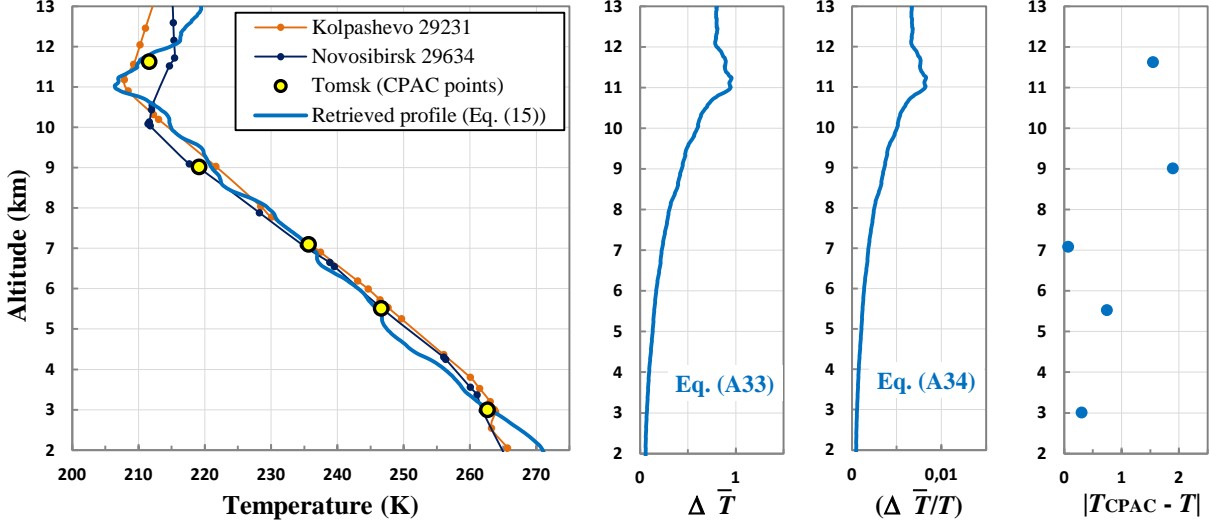

**Figure 8.** (1 April 2015) Temperature profile retrieved using the temperature retrieval function (Eq. 15) derived from the calibration function suggested by Gerasimov and Zuev (2016). The uncertainties $\Delta \overline{T}$ and $(\overline{\Delta T / T})$ are calculated by Eqs. (A33) and (A34), respectively.




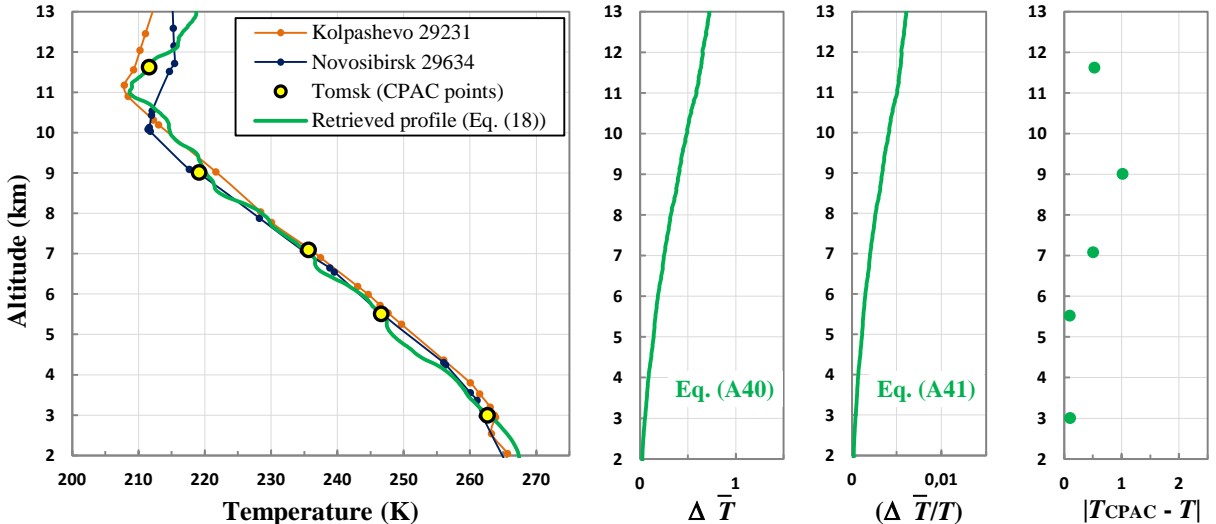

**Figure 9.** (1 April 2015) Temperature profile retrieved using the temperature retrieval function (Eq. 18) derived from the calibration function suggested by Lee III (2013). The uncertainties $\Delta\overline{T}$ and $(\overline{\Delta T / T})$ are calculated by Eqs. (A40) and (A41), respectively.

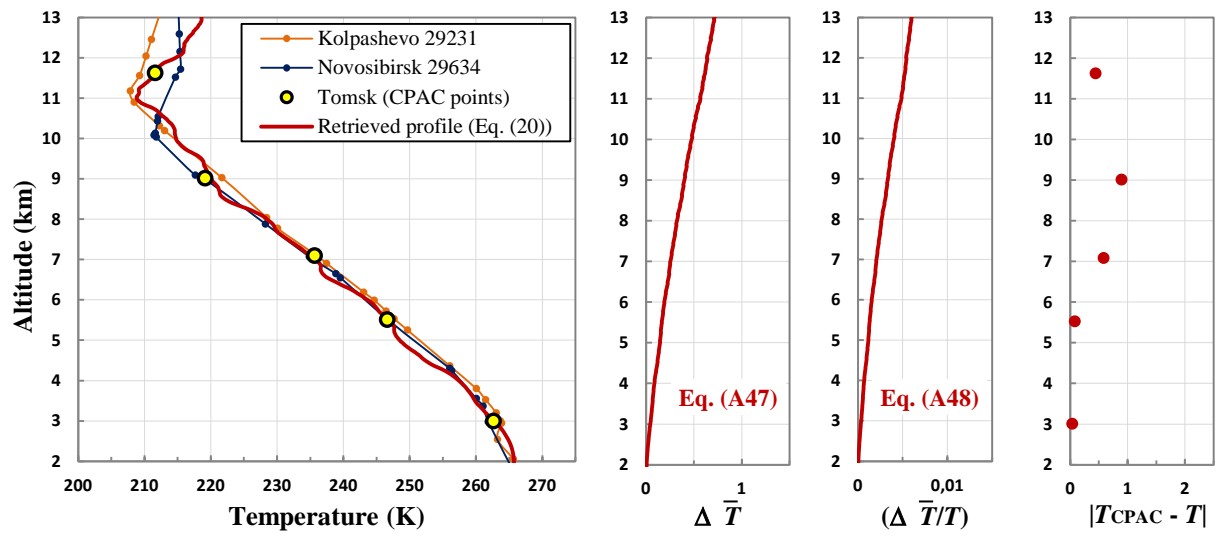

**Figure 10.** (1 April 2015) Temperature profile retrieved using the temperature retrieval function (Eq. 20) derived from the calibration function suggested by Gerasimov and Zuev (2016). The uncertainties $\Delta\overline{T}$ and $(\overline{\Delta T / T})$ are calculated by Eqs. (A47) and (A48),

10    respectively.





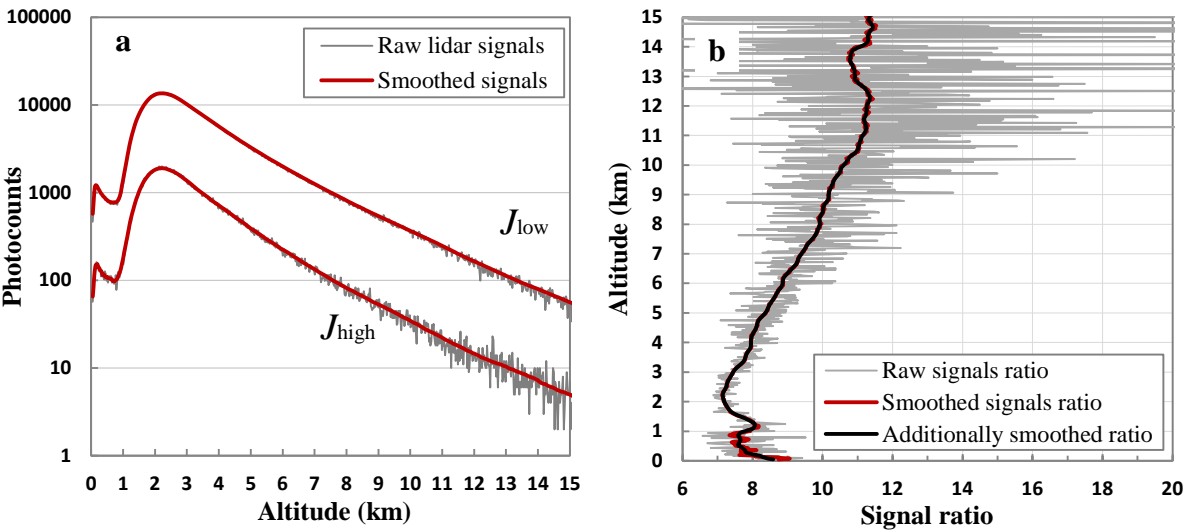

**Figure 11.** IMCES lidar data taken between 20:21 and 21:21 LT on 2 October 2014 (13:21–14:21 UTC). **(a)** Raw photocounts $N_L$ and $N_H$ detected in the lidar channels with $J_{low}$ and $J_{high}$, respectively, together with the smoothed ones $\overline{N}_L$ and $\overline{N}_H$. **(b)** Raw photocounts ratio $Q$ = $N_L/N_H$, smoothed photocounts ratio $Q = \overline{N}_L/\overline{N}_H$, and additionally smoothed ratio.

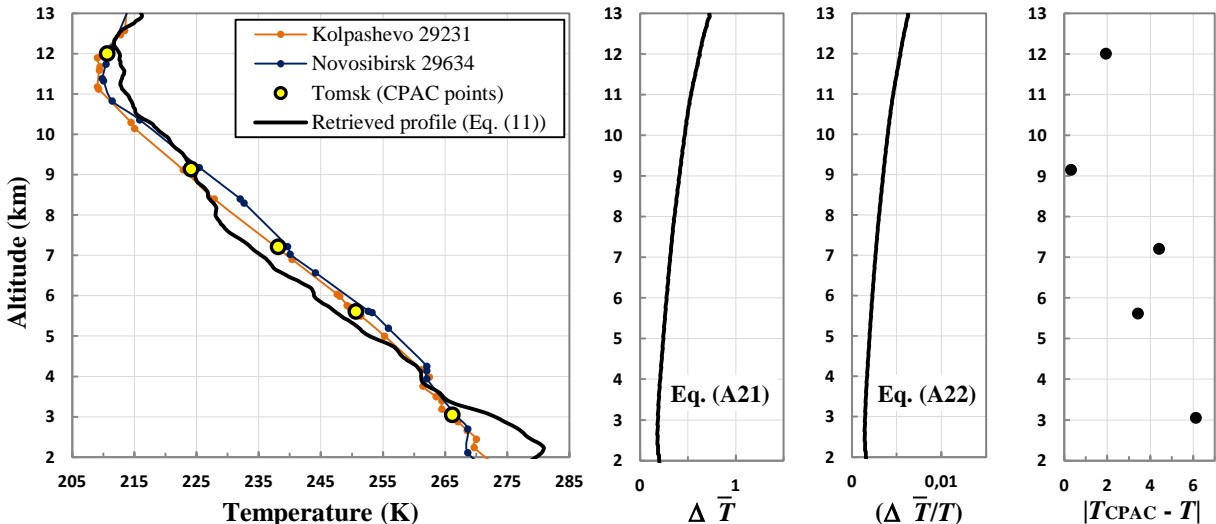

**Figure 12.** (2 October 2014) Temperature profile retrieved using Eq. (11). The absolute and relative uncertainties $\Delta\overline{T}$ and $(\overline{\Delta T}/T)$ are calculated by Eqs. (A21) and (A22), respectively.





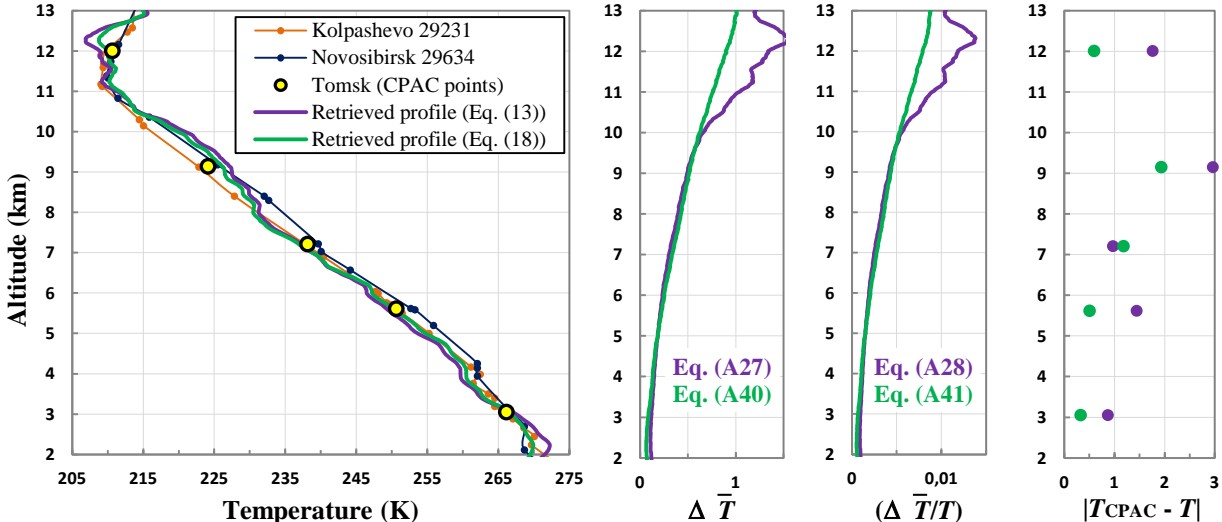

**Figure 13.** (2 October 2014) Temperature profiles retrieved using Eqs. (13) and (18).

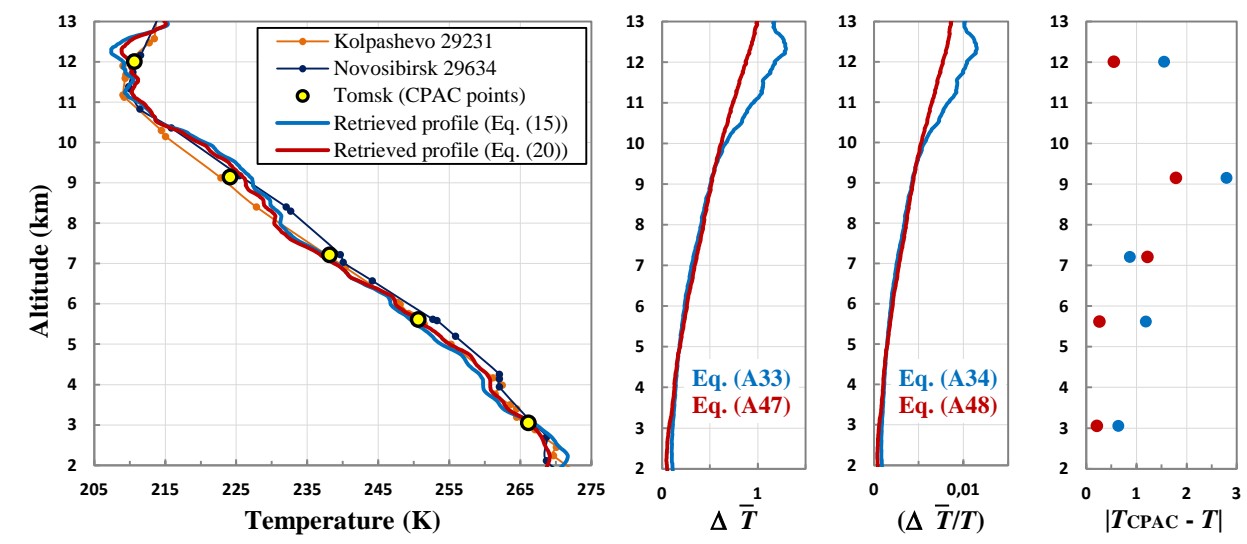

**Figure 14.** (2 October 2014) Temperature profiles retrieved using Eqs. (15) and (20).