# Peer review of "Tropospheric temperature retrievals using nonlinear calibration functions in the pure rotational Raman lidar technique"

_Atmospheric Measurement Techniques, 2016_

## Referee Comment (RC1) · Anonymous Referee #1 · 24 Aug 2016

This paper presents a methodology to derive a nonlinear calibration curve for the relation between the observed signals and the temperature in the pure rotational Raman technique. The classical linear calibration curve is exact only if we use 2 single rotational Raman lines. In practice most lidar systems select 2 spectral bands with several Raman lines in each band and the linear calibration curve is no more exact. It is then valuable to try to improve the results taking into account nonlinear terms in the calibration curve. The application of nonlinear calibration curves to true lidar data show that it improves the accuracy of the comparison with external temperature data. However they are some weaknesses in the computation of uncertainties developed in Appendices A0 to A3 that makes the paper unpublishable in its present state and I recommend a major

revision. The reasons for this recommendation are explained below.

The Formula (A2) giving the uncertainty on the ratio Q between the two Raman lidar channels is not correct. As the signals in the two channels are independent of each other, the uncertainties should not be summed linearly but quadratically. The derivation of $\Delta T$ using Formulas (A3) to (A8) is therefore also not correct. Surprisingly the Formula A9 giving the uncertainty on $\Delta T/T$ is correct but I don't understand how it is possible to derive it from (A8). As a consequence there is an inconsistency in the experimental results on $\Delta T$ and $\Delta T/T$ presented on Figures 6 to 10 and 12 to 14. The ratio between $\Delta T$ and $\Delta T/T$ should be equal to the temperature T that varies between 270 K and 205 K in the altitude range covered by the lidar. The ratio on the Figures seems to be more in the order of 120K, with for instance $\Delta T/T=0.005$ and $\Delta T=0.6K$. The same mistake exists also in Appendices A1 to A3.

Concerning the experimental results, the estimation of the temperature difference with the reference data CPAC is not affected by the uncertainty computation and can be considered as valid. It is clear that a nonlinear curve gives globally better results than the linear curve. It is especially true in the lower part of the atmosphere. However it is not so clear that it improves also the results above 8 km. In some cases the linear curve gives better results that the nonlinear ones, for instance at 9 km. Do the authors have an explanation for that and is it necessary to apply a nonlinear curve in the full tropospheric range or only in the lower part?

---

## Referee Comment (RC2) · Anonymous Referee #2 · 4 Sep 2016

**General comments:

The authors investigate different mathematical equations for the calibration of tropospheric (and lower stratospheric) temperature measurements with rotational Raman lidar (RRL). The topic of RRL calibration has already been discussed extensively in the literature (see references given also in the manuscript). It has been shown both within simulations and a very large set of experimental data taken with several different RRL systems that the systematic deviations between RRL measurements and collocated radiosounding data are small. Differences in individual cases can be attributed to the well-understood statistical uncertainties of the RRL measurements (which increase with range of the lidar data) or sampling of different air as well different weighting func-

tions of the data (which fluctuate and average out for a large number of comparisons). The general statement that the "commonly used calibration function" would yield significant errors of 1 K is wrong (abstract, line 14). The bias of current state-of-the-art RRL systems is only < 0.5 K (see Wulfmeyer et al. 2015 for a recent review).

The authors claim that their work goes beyond previous publications because they take the collisional broadening of the lines within the pure rotational Raman backscatter spectrum into account. While the first and second author of this manuscript have recently published simulations on this topic, the goal of this manuscript is to show experimental comparisons between RRL temperature data obtained with different calibration functions and "reference" data. Unfortunately, the study lacks such suitable reference data. The RRL measurements were taken at a site for which unfortunately no local radiosonde ascents were available. The closest radiosonde launching sites are more than 250 km away (see section 4.2). But this is clearly much too far for suitable comparisons. In addition, the authors use data of constant pressure altitude charts with low vertical resolution to overcome the large distances to the radiosonde sites but also these are certainly related to too large uncertainties for being used as reference data for investigating the small differences between the calibration functions. Also with a perfect calibration function, one could not "force" uncorrelated temperature data to agree.

It may be possible that the lidar system used by the authors is special and that for this system a more complicated calibration function is needed than for other RRL systems described in the literature. The calibration errors depend of the spectral characteristics of the lidar receiver, namely the widths, central wavelengths, shapes of the transmission functions as well as whether just the anti-Stokes or both branches of the pure rotational Raman spectrum are collected. I could imagine that especially the last point combined with narrow transmission bands may lead to larger calibration errors when using a too simple calibration function. Maybe this could explain why the commonly used calibration functions do not seem to work well for the RRL discussed here. More

simulations for different types of RRL systems would be needed to verify this hypothesis – or, even better, collocated RRL measurements with different types of systems. Maybe this very interesting experiment could be realized in the future.

My suggestion is to revise the manuscript substantially and to rewrite the statements which are too general by clarifying that this study is on the data of a special/unique RRL system and comparisons with model temperature data which possess certain uncertainties. It should be made clear that the results may be used as indication (not proof) that some RRL systems may require more complicated calibration functions than the ones reported so far in the literature and that better reference data for comparisons will be useful to support this interpretation.

\*\*Specific comments:

Title: Maybe better "Tropospheric temperature measurements with the pure rotational Raman lidar technique using nonlinear calibration functions". RRL does not "retrieve" the temperature; there is no first guess like in passive remote sensing.

Abstract, line 13 ff: This general statement is (fortunately) wrong. The calibration functions used so far lead (for all systems discussed in the literature) much smaller errors due to the calibration function itself. In addition, the error depends on the temperature range of the calibration. See general comments above. Please rewrite.

Abstract, line 15 ff: The statement that "collisional broadening... cannot be neglected (for) tropospheric temperature measurements" is too general. Again, this depends on the individual characteristics of the RRL system and the temperature range etc.. Please omit or clarify.

Page 2, line 4 and section 4.1: I do not like the term "smoothing" because it could include different types of filters which are not preferable and not meant. I suggest simply writing "averaging in time and range". Further averaging of the ratio should be avoided. It only complicates the effective weighting function of the resulting data while

the averaging of the raw data should anyhow be made with sufficiently large windows in order to avoid too large noise errors when taking the ratio.

Equations 1, 2, 3 and related text: The instrumental efficiency which is different for different lines is not yet included here but should be included. Otherwise, "calibration" does not make sense.

Section 4.1: What is the resolution of the raw data? As said above, further averaging of the ratio is not preferable. What is otherwise the effective weighting function of the double-averaged data?

Section 4.3: It should be made clear that the CPAC data are not a reference. Thus large differences between the RRL and CPAC data are not necessarily due to problems with the calibration function. I suggest that you show in addition the calibration plots (T_CPAC versus T_RRL with calibration function).

Appendix A: The propagation of the Poisson errors have already been discussed extensively in the literature for the calibration functions used so far – also including the contribution of the background signal which is missing here. Thus, these parts should be deleted here. Instead, references to the existing literature should be given which are currently missing.

Figure 1: Please explain also the red and blue curves and identify the laser wavelength.

Figure 2: Should be deleted as this photo does not explain any technical details. Figure 3 is enough and much better I think.

Figure 3: Which PMT is used for which signal?

**Reference:

Wulfmeyer, V., R. M. Hardesty, D. D. Turner, A. Behrendt, M. P. Cadeddu, P. Di Girolamo, P. Schlüssel, J. Van Baelen, and F. Zus, 2015: A review of the remote sensing of lower-tropospheric thermodynamic profiles and its indispensable role for the understanding and the simulation of water and energy cycles. Rev. Geophys. 53 (3), 819–895. DOI:10.1002/2014RG000476

---

## Author Comment (AC1) · 4 Nov 2016

Manuscript Number: amt-2016-189

Manuscript Type: research article

**Title:** Tropospheric temperature retrievals using nonlinear calibration functions in the pure rotational Raman lidar technique

First of all, we want to thank both Reviewers for their valuable criticism, comments, and suggestions which allow to improve our manuscript.

**Point-by-point response to Referee 1**

**Comment:** ... However they are some weaknesses in the computation of uncertainties developed in Appendices A0 to A3 that makes the paper unpublishable in its present state and I recommend a major revision. The reasons for this recommendation are explained below. The Formula (A2) giving the uncertainty on the ratio Q between the two Raman lidar channels is not correct. As the signals in the two channels are independent of each other, the uncertainties should not be summed linearly but quadratically. The derivation of  $\Delta T$  using Formulas (A3) to (A8) is therefore also not correct. Surprisingly the Formula A9 giving the uncertainty on  $\Delta T/T$  is correct but I don't understand how it is possible to derive it from (A8). As a consequence there is an inconsistency in the experimental results on  $\Delta T$  and  $\Delta T/T$  presented on Figures 6 to 10 and 12 to 14. The ratio between  $\Delta T$  and  $\Delta T/T$  should be equal to the temperature T that varies between 270 K and 205 K in the altitude range covered by the lidar. The ratio on the Figures seems to be more in the order of 120K, with for instance  $\Delta T/T = 0.005$  and  $\Delta T = 0.6$  K. The same mistake exists also in Appendices A1 to A3.

**Our response:** We agree with Referee's notes regarding the mentioned formulas and Appendices. We have revised and rewritten all Appendices (A and A0–A4), and corrected equations, figures, and the Supplement data. See, please, the corrected Appendices in the list of corrections or in the revised blue-colored manuscript below.

**Comment:** Concerning the experimental results, the estimation of the temperature difference with the reference data CPAC is not affected by the uncertainty computation and can be considered as valid. It is clear that a nonlinear curve gives globally better results than the linear curve. It is especially true in the lower part of the atmosphere. However it is not so clear that it improves also the results above 8 km. In some cases the linear curve gives better results that the nonlinear ones, for instance at 9 km.

**Response:** The linear (black) curve is better than nonlinear ones (colored) only at one point (9 km). This is a random result due to a small number of the "reference" CPAC points. As seen from the comparative analysis of the difference  $|T_{CPAC} - T|$  between temperature values retrieved from the CPACs and IMCES lidar data in Figure 1 (below), three calibration functions (red, green, and blue curves) retrieve the temperature better than the linear one at an altitude of ~11.5 km.

Figure 1. Difference  $|T_{CPAC} - T|$  between temperature values retrieved from the CPACs and IMCES lidar data.

**Comment:** Do the authors have an explanation for that and is it necessary to apply a nonlinear curve in the full tropospheric range or only in the lower part?

**Response**: As we have experimentally shown in Figure 1 and in our previous Optics Express paper via simulation (Gerasimov and Zuev, 2016), the considered nonlinear calibration functions are preferable for temperature retrievals in the full tropospheric range.

Gerasimov, V. V. and Zuev, V. V.: Analytical calibration functions for the pure rotational Raman lidar technique, Opt. Express, 24, 5136–5151, 2016.

**Point-by-point response to Referee 2**

**Comment:** ... The general statement that the "commonly used calibration function" would yield significant errors of 1 K is wrong (abstract, line 14). The bias of current state-of-the-art RRL systems is only

Fig. 10.2. (a) Typical intensities of the two pure-rotational Raman signals  $S_{RR1}$  and  $S_{RR2}$  as a function of temperature T [48]. (b) Signal ratio Q from which the atmospheric temperature is derived.  $S_{ref}$  can be used as a temperature-independent Raman reference signal for measuring extinction and backscatter coefficients of aerosols and cloud particles [47].

state-of-the-art radiosondes are accurate within tenths of a K provided the radiosonde itself has been accurately calibrated. Of course, the reference data used for the calibration should be taken as close in space and time as possible to the atmospheric column sensed by the lidar. How often an RR lidar systems needs to be recalibrated depends on the individual system. Provided that rugged mounts are used and the alignment of the lidar is not changed intentionally, the calibration of today's state-of-art systems remains virtually unchanged and only long-term degradations of the optical components may require recalibrations on a longer time scale.

For systems that detect only one RR line in each of the two RR channels Eq. (10.25) takes the simple form

$$Q(T) = \exp(a - b/T),$$
 (10.26)

where the parameters *a* and *b* are both positive if  $J(S_{RR2}) > J(S_{RR1})$ . *b* is simply the difference of the rotational Raman energies of the extracted lines divided by *k*, and *a* is the logarithm of the ratio of all factors except the exponential term in Eq. (10.20). It is straightforward to use Eq. (10.26) also for systems with several lines in each of the RR signals [42]. But the obvious inversion of Eq. (10.26) which gives

$$T = \frac{b}{a - \ln Q} \tag{10.27}$$

then turns out to yield significant measurement errors, well in excess and the approach of Eq. (10.28), which both require four calibration conof 1 K (cf. Fig. 10.3) when measurements are made over an extended stants. For Eq. (10.29), the temperature derived from the data with that

**Instead of**

"The commonly used calibration function linear in reciprocal temperature ignores the broadening of individual atmospheric  $N_2$  and  $O_2$  PRR lines and, at the same time, yields significant errors (±1 K) in temperature retrievals."

Fig. 10.3. Errors made with different calibration functions for rotational Raman temperature lidar.

range of temperatures. As the errors behave nearly as a second-orderpolynomial function of temperature, it has been proposed to minimize calibration errors by a second calibration with such a second-order polynomial [43, 62], leading to a calibration function of the form

$$T = \frac{b}{a - \ln Q} + c \left(\frac{b}{a - \ln Q}\right)^2 + d \qquad (10.28)$$

with the additional calibration constants c and d.

An even better calibration function, however, is found in the approach

$$Q = \exp\left(\frac{a'}{T^2} + \frac{b'}{T} + c'\right) \Longleftrightarrow T = \frac{-2a'}{b' \pm \sqrt{b'^2 - 4a'(c' - \ln \mathbf{Q})}},$$
(10.29)

which extends Eq. (10.26) to a second-order term in *T* and needs only three calibration constants
$$a', b', c'$$
. Fitting, as an example, the curve  $T(Q)$  shown in Fig. 10.2 with the different calibration functions, one gets the calibration errors shown in Fig. 10.3. The performances of polynomial calibration functions are also given for comparison. The single-line approach of Eq. (10.27) results here in errors of  $\sim \pm 1 \text{ K}$  for temperatures between 180 and 285 K, which is better than a linear calibration function. However, this relation is not generally valid [63]. For three calibration constants, Eq. (10.29) is superior to the second-order polynomial and even better than the third-order polynomial and the approach of Eq. (10.28), which both require four calibration constants.

**we wrote**

"...The commonly used calibration function (linear in reciprocal temperature 1/T with two calibration coefficients) ignores all types of broadening of individual PRR lines of atmospheric N2 and O2 molecules." [Page 1, lines 13–14, revised manuscript]

**Comment:** ... While the first and second author of this manuscript have recently published simulations on this topic, the goal of this manuscript is to show experimental comparisons between RRL temperature data obtained with different calibration functions and "reference" data. Unfortunately, the study lacks such suitable reference data. The RRL measurements were taken at a site for which unfortunately no local radiosonde ascents were available. The closest radiosonde launching sites are more than 250 km away (see section 4.2). But this is clearly much too far for suitable comparisons. In addition, the authors use data of constant pressure altitude charts with low vertical resolution to overcome the large distances to the radiosonde sites but also these are certainly related to too large uncertainties for being used as reference data for investigating the small differences between the calibration functions. Also with a perfect calibration function, one could not "force" uncorrelated temperature data to agree.

**Response**: We agree with this Referee's note about "reference" data. However, the temperature points retrieved using available constant pressure altitude charts (CPACs) with the temperature accuracy of 0.5 K and the vertical accuracy of 20 m allow to make the comparative analysis of temperature uncertainties, yielded by using different calibration functions, and determine the best-suited function for our lidar system. So, we added two explaining sentences at the end of Sect. 4.2.

**Instead of**

"...the University of Wyoming (Novosibirsk and Kolpashevo station numbers are 29634 and 29231, respectively)."

**we wrote**

"...the University of Wyoming (Novosibirsk and Kolpashevo station numbers are 29634 and 29231, respectively). It is clear that the CPAC points are not suitable for using them as the reference points to calibrate lidars and retrieve temperature profiles with high accuracy (for this purpose the local radiosonde data are required). Nevertheless, the accuracy of these points (0.5 K, 20 m) is sufficient to make the comparative analysis of temperature uncertainties, yielded by using different calibration functions, and determine the best-suited function (among them) for our lidar system."

[Page 8, lines 17–22, revised manuscript]

**Comment**: It may be possible that the lidar system used by the authors is special and that for this system a more complicated calibration function is needed than for other RRL systems described in the literature. The calibration errors depend of the spectral characteristics of the lidar receiver, namely the widths, central wavelengths, shapes of the transmission functions as well as whether just the anti-Stokes or both branches of the pure rotational Raman spectrum are collected. I could imagine that especially the last point combined with narrow transmission bands may lead to larger calibration errors when using a too simple calibration function. Maybe this could explain why the commonly used calibration functions do not seem to work well for the RRL discussed here. More simulations for different types of RRL systems would be needed to verify this hypothesis – or, even better, collocated RRL measurements with different types of systems. Maybe this very interesting experiment could be realized in the future.

**Response**: We agree. That's quite possible. Despite the nonlinear calibrations functions, derived in (Gerasimov and Zuev, 2016) and applied in the current AMTD paper, represent a direct consequence of the collisional broadening of PRR lines, we do not exclude the possibility that the best-suited function can depend on the PRR lidar system. We noted that at the end of Sect. 6.

**Instead of**

"As the best function for temperature retrievals can depend on a lidar system (e.g., based on DGs or IFs for PRR lines extracting), it is reasonable to check all the mentioned nonlinear functions against lidar data obtained with different lidar systems to determine the best function in each specific case. As the collisional broadening of PRR lines is the largest in the atmospheric boundary layer, the nonlinear calibrations functions should be applied instead of the linear one for temperature retrievals, especially if using a coaxial lidar."

**we wrote**

As it was mentioned above (Sect. 4.2), the CPAC points can hardly be used as the reference data to reliably calibrate PRR lidars and retrieve accurate temperature profiles. Nevertheless, the results suggest that the best-suited calibration function for temperature retrievals can depend on the lidar system (e.g., based on DGs or IFs for PRR lines extracting), which can take into account the collisional broadening of PRR lines in varying degrees. Indeed, the calibration errors depend on the spectral characteristics of the lidar receiver such as the central wavelength, shape and width of the transmission functions, as well as whether just the anti-Stokes (IFs) or both branches of the PRR spectrum (DGs) are used to extract the PRR signals from backscattered light. Therefore, it is reasonable to check all the mentioned nonlinear functions against lidar data obtained with different lidar systems to determine the best function in each specific case." [Page 10, lines 26–31 and Page 11, lines 1 and 2, revised manuscript]

**Comment**: My suggestion is to revise the manuscript substantially and to rewrite the statements which are too general by clarifying that this study is on the data of a special/unique RRL system and comparisons with model temperature data which possess certain uncertainties. It should be made clear that the results may be used as indication (not proof) that some RRL systems may require more complicated calibration functions than the ones reported so far in the literature and that better reference data for comparisons will be useful to support this interpretation.

**Response**: We have deeply revised the text of our manuscript. The abstract, Sects. 4.1 and 6, and Appendix A were significantly rewritten (see, please, the list of corrections or revised blue-colored manuscript below). The results can be considered as preliminary in the absence of reliable radiosonde data, and we offer to lidar researcher to use these nonlinear functions and determine the best one in each specific case of the PRR lidar system.

**Specific comments of Referee 2**

**Comment:** Title: Maybe better "Tropospheric temperature measurements with the pure rotational Raman lidar technique using nonlinear calibration functions". RRL does not "retrieve" the temperature; there is no first guess like in passive remote sensing.

**Response**: Thank you for your suggestion. Yes, we know that calibration function retrieve the temperature, not a PRR lidar. To make the title more clear, we slightly rewrote it.

**Instead of**

"Tropospheric temperature retrievals using nonlinear calibration functions in the pure rotational Raman lidar technique"

**we wrote**

"Tropospheric temperature retrievals using nonlinear calibration functions in the frame of the pure rotational Raman lidar technique."

[Page 1, lines 1 and 2, revised manuscript]

**Comment:** Abstract, line 13 ff: This general statement is (fortunately) wrong. The calibration functions used so far lead (for all systems discussed in the literature) much smaller errors due to the calibration function itself. In addition, the error depends on the temperature range of the calibration. See general comments above. Please rewrite.

**Response:** It was already discussed above.

**Comment:** Abstract, line 15 ff: The statement that "collisional broadening ... cannot be neglected (for) tropospheric temperature measurements" is too general. Again, this depends on the individual characteristics of the RRL system and the temperature range etc. Please omit or clarify.

**Response: Rewritten.**

**Instead of**

"However, the collisional (or pressure) broadening of  $N_2$  and  $O_2$  PRR lines dominates over other types of broadening in the troposphere, and therefore, cannot be neglected during tropospheric temperature measurements."

**we wrote**

"However, the collisional (pressure) broadening dominates over other types of broadening of PRR lines in the troposphere and can differently affect the accuracy of tropospheric temperature measurements depending on the PRR lidar system."

[Page 1, lines 13–15, revised manuscript]

**Comment:** Page 2, line 4 and section 4.1: I do not like the term "smoothing" because it could include different types of filters which are not preferable and not meant. I suggest simply writing "averaging in time and range".

**Response:** The term "smoothing" was substituted by the term "averaging" throughout the manuscript.

**Comment:** Further averaging of the ratio should be avoided. It only complicates the effective weighting function of the resulting data while the averaging of the raw data should anyhow be made with sufficiently large windows in order to avoid too large noise errors when taking the ratio.

**Response:** We have to disagree with this suggestion. In some cases, the second-order averaging of raw data (or/and their ratio) is required and more preferable than the first-order one (see, e.g., El'nikov et al., 2000). Here we applied the second-order averaging of raw data to reduce signal statistical fluctuations (see Sect. 4.1, Appendix A, and examples below). Both the uncertainties  $\Delta \overline{T}$  and  $(\overline{\Delta T}/T)$  decrease by  $\sqrt{n} = \sqrt{11}$  times, when using additional slight smoothing for our lidar data. For example:

**(1 April 2015, Eq. 13)**